# Single-cell genomics improves the discovery of risk variants and genes of atrial fibrillation

Alan Selewa[1,7], Kaixuan Luo[2,7], Michael Wasney ®[3], Linsin Smith[4], Xiaotong Sun[2], Chenwei Tang[5], Heather Eckart ®[3], Ivan P. Moskowitz ®[2,6], Anindita Basu[3] ✉, Xin He ®[2,8] ✉ & Sebastian Pott ®[3,8] ✉

Genome-wide association studies (GWAS) have linked hundreds of loci to cardiac diseases. However, in most loci the causal variants and their target genes remain unknown. We developed a combined experimental and analytical approach that integrates single cell epigenomics with GWAS to prioritize risk variants and genes. We profiled accessible chromatin in single cells obtained from human hearts and leveraged the data to study genetics of Atrial Fibrillation (AF), the most common cardiac arrhythmia. Enrichment analysis of AF risk variants using cell-type-resolved open chromatin regions (OCRs) implicated cardiomyocytes as the main mediator of AF risk. We then performed statistical fine-mapping, leveraging the information in OCRs, and identified putative causal variants in 122 AF-associated loci. Taking advantage of the fine-mapping results, our novel statistical procedure for gene discovery prioritized 46 high-confidence risk genes, highlighting transcription factors and signal transduction pathways important for heart development. In summary, our analysis provides a comprehensive map of AF risk variants and genes, and a general framework to integrate single-cell genomics with genetic studies of complex traits.

Cardiac diseases are a leading cause of mortality across the world[1,2]. GWAS of cardiac traits have uncovered a large number of associations, including more than 100 loci linked to atrial fibrillation (AF)[3–7]. However, in most loci the disease-driving causal variants remain unknown. Given that most trait-associated variants are located in non-coding regions[8], researchers often use regulatory and epigenomic datasets to annotate possible effects of variants, and to prioritize putative causal variants[8–10]. Existing datasets, however, were often collected from bulk tissue samples that represent complex mixtures of cell types[11,12], while disease-causing variants often act in specific cell types. Thus, lack of cell type-resolved epigenomic data in disease-related tissues limits our ability of variant annotation and prioritization. Another challenge of post-GWAS analysis is that

long-range gene regulation is common, making it difficult to link non-coding variants with their target genes.

Despite these challenges, researchers have made attempts to identify putative risk variants and genes underlying the AF genetics. One study used epigenomic and gene expression data in the human heart to nominate putative risk genes in 104 AF-associated loci. This study, however, did not use rigorous statistical analysis to fine-map causal variants and instead used relatively lenient cutoffs and an ad hoc scoring scheme to rank putative target genes[13]. This study nominated nearly 300 genes in these loci, many of which are likely not causal genes. Another study used STARR-seq to map regulatory regions and variants to nominate risk variants in 12 AF-associated loci[14]. But the majority of AF-associated loci were not investigated in the study.

[1]Biophysical Sciences Graduate Program, The University of Chicago, Chicago, IL 60637, USA. [2]Department of Human Genetics, The University of Chicago, Chicago, IL 60637, USA. [3]Department of Medicine, Section of Genetic Medicine, The University of Chicago, Chicago, IL 60637, USA. [4]Committee on Genetics, Genomics and Systems Biology, The University of Chicago, Chicago, IL 60637, USA. [5]The College, The University of Chicago, Chicago, IL 60637, USA. [6]Department of Pediatrics, The University of Chicago, Chicago, IL 60637, USA. [7]These authors contributed equally: Alan Selewa, Kaixuan Luo. [8]These authors jointly supervised this work: Xin He, Sebastian Pott. ✉e-mail: obasu@uchicago.edu; xinhe@uchicago.edu; spott@uchicago.edu

A more recent study collected single-cell RNA-seq and ATAC-seq data in the human heart, and performed fine-mapping in AF-associated loci[15]. The study identified 38 putative risk variants in heart *cis-regulatory elements* (CREs). Nevertheless, few nominated variants reach high confidence and in most loci the risk genes remain unknown.

To address these challenges in the context of heart diseases, we developed an integrated framework that unifies advances in single cell epigenomics, computational fine-mapping and a novel procedure for risk gene discovery. Specifically, we performed single-cell chromatin accessibility profiling to map open chromatin regions (OCRs) across major cell types in the heart. Our statistical fine-mapping method utilizes these candidate CREs to infer disease-relevant cell types and takes advantage of this information to identify putative causal variants. Our novel gene-mapping approach then aggregates information of all fine-mapped SNPs to predict the risk genes, considering multiple sources of information such as distance and chromatin loops between enhancers and promoters. Application of this framework to AF revealed a number of putative risk variants and genes, highlighting biological processes important to the genetics of AF.

An unexpected finding from our study is that the majority of risk variants of AF we discovered did not colocalize with heart eQTLs. Taking advantage of our cell-type resolved epigenomic data, we found that this was largely due to the lack of power of bulk eQTL studies to identify regulatory variants with cell-type specific effects. This finding thus sheds light on the common strategy of annotating GWAS results using eQTLs.

## Results

### Overview of the experimental and computational approach

Our approach combines single-cell genomics with novel computational procedures to study genetics of cardiac traits (Fig. 1). Using single nucleus RNA-sequencing[16–18] (snRNA-seq) and single-cell ATAC-seq (scATAC-seq)[19,20], we obtained transcriptome and open chromatin regions (OCRs) across all major cell types in the adult human heart (Fig. 1, step 1). These OCR profiles allow us to discover cell types enriched with the genetic risks of traits of interest. To identify specific causal variants in trait-associated loci, we performed Bayesian statistical fine-mapping. Fine-mapping is a technique that aims to identify one or few causal variants that explain all the associations in a locus. It avoids the use of arbitrary LD cutoffs in selecting candidate variants and is able to quantify the uncertainty of each nominated variant. Recent fine-mapping techniques are also able to incorporate functional information of variants, such as regulatory activities in trait-related cell types[9,21,22]. Because of these benefits, fine-mapping techniques have been successfully applied to many common traits such as Type 2 Diabetes[23], Schizophrenia[24] and autoimmune disorders[25]. Our fine-mapping method takes advantage of the cell-type-resolved chromatin data to favor variants located in OCRs of enriched cell types (Fig. 1, step 2). After fine-mapping, the candidate SNPs and their associated cell-type information allow us to assign the cell type(s) through which the causal variants are likely to act.

Finally, we developed a procedure to infer causal genes at each locus (Fig. 1, step 3), addressing some common challenges. In "gene

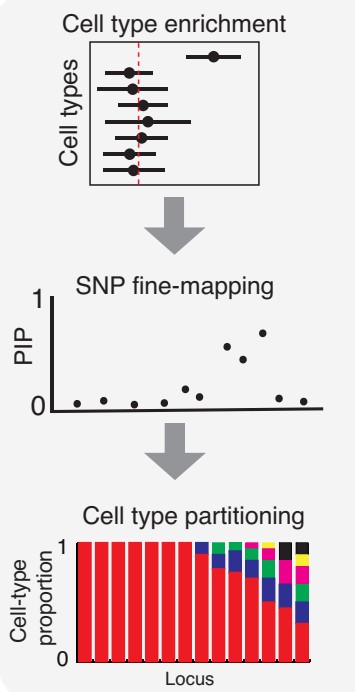
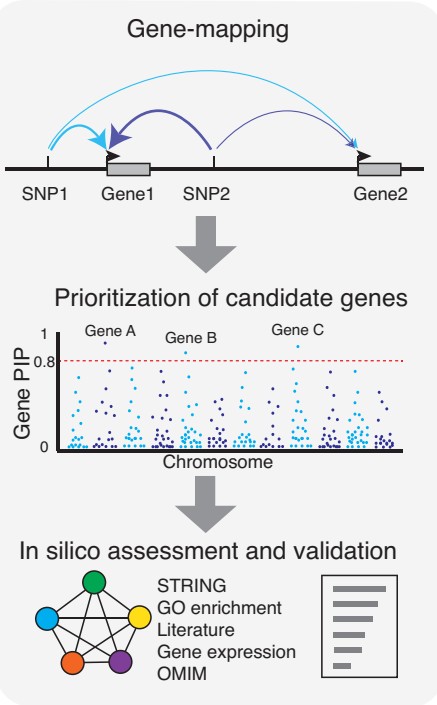

**Fig. 1 | Overview of our experimental and computational framework.** Left: snRNA-seq and scATAC-seq profiling to cluster cells and obtain open chromatin regions (OCRs) in each cell type. Schematic of the heart is provided by OpenClipart-Vectors via Pixabay. Middle: Using OCRs and GWAS summary statistics to assess variant enrichment in cell-type-resolved OCRs. The enrichment results then provide prior for Bayesian statistical fine-mapping. The resulting Posterior Inclusion Probabilities (PIPs) represent the probabilities of variants being causal. The likely cell types through which the causal signals at each locus act can be identified by considering cell type information of likely causal variants. We may not always be able to identify a single cell type per locus, so we assign probabilities to cell types. Right: Computational gene-mapping using PIPs from SNP fine-mapping and SNP-to-gene links to obtain gene level PIPs. Note that the PIP of a SNP is partitioned into nearby genes in a weighted fashion, with more likely target genes receiving higher weights (as indicated by thicker arrows). Prioritized genes can be further assessed through external evidence such as gene networks and expression.

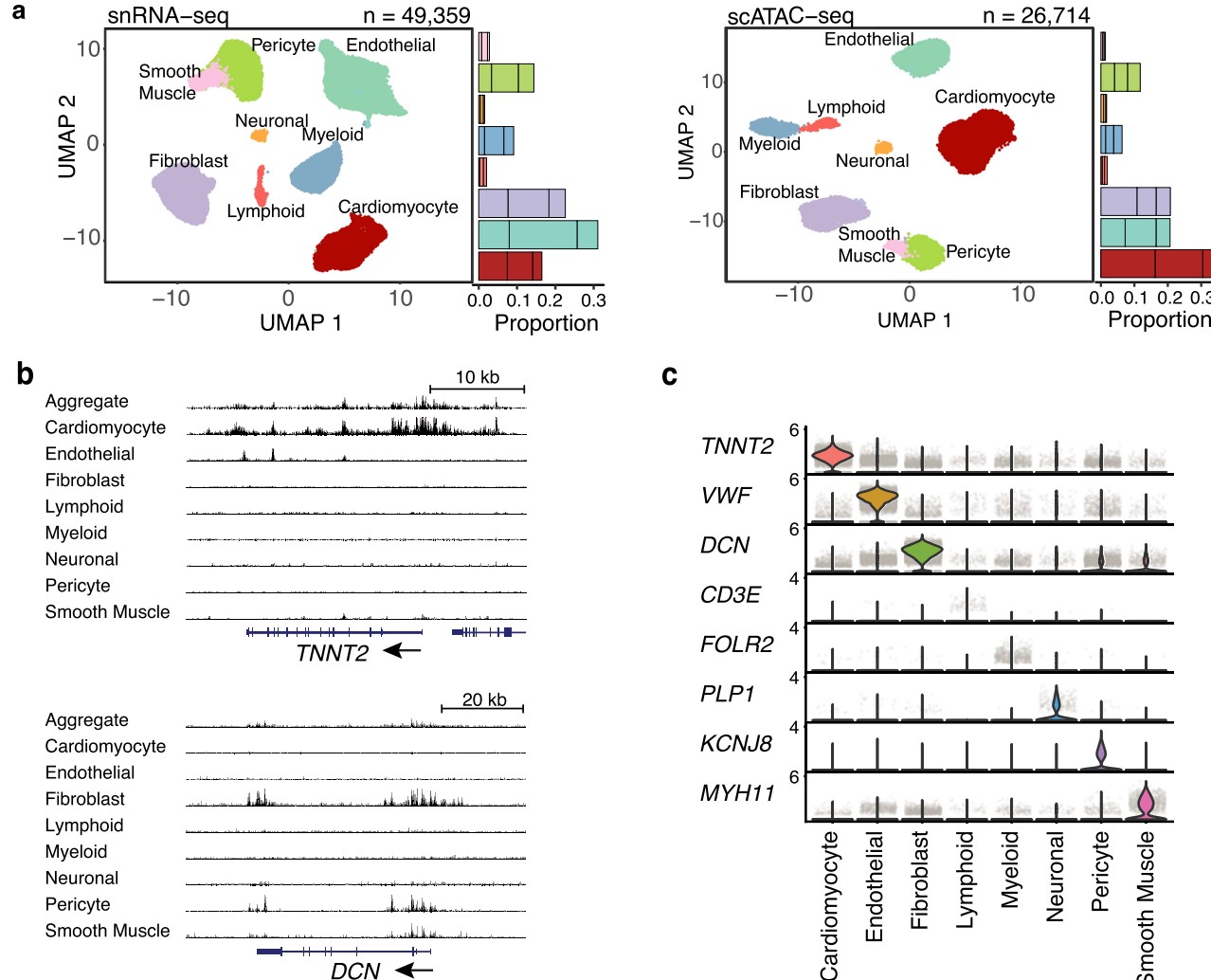

**Fig. 2 | Mapping cell types in the human heart. a** UMAP projection of individual cells from snRNA-seq and scATAC-seq colored by cell types. Stacked barplots on the right represent the proportions of cell-types from each of the three donors. **b** UCSC genome browser track plots of chromatin accessibility at selected marker genes across cell types. The bottom part shows the gene track (RefSeq annotation). Shown are two marker genes, *TNNT2*: cardiomyocyte marker; *DCN*: fibroblast marker. **c** Stacked violin plots of marker gene expression (log-normalized expression values) in each cell type.

association tests" researchers test if the set of SNPs near a gene collectively show disease association[26,27]. These types of methods, however, cannot distinguish between multiple genes close to disease-associated variants. Alternatively, researchers may perform fine-mapping first, then link the high-confidence SNPs to target genes using additional information. However, fine-mapping alone rarely leads to a single, or even a few, high confidence SNPs at associated loci[23], therefore this approach also has limited utility. In contrast, our procedure aggregates information of all fine-mapped variants in a locus to nominate risk genes. To see its benefit, suppose fine-mapping in a locus implicates 10 putative causal variants without any single one reaching high confidence; however, if all 10 SNPs likely target the same gene, we can be confident of the causal gene. We developed a statistical procedure to implement this idea, and estimated a score, called gene PIP (Posterior Inclusion Probability), for each gene. Under certain assumptions, we showed that the gene PIP estimates the probability of a gene being causal. The details are described below and in "Methods".

**Single-cell transcriptome and chromatin accessibility profiling reveals multiple cell types in the human heart**

We performed snRNA-seq and scATAC-seq using the Chromium platform (10x Genomics) (Fig. 1, step 1). The heart samples were obtained from the left and right ventricles (LV and RV), the interventricular septum, and the apex of three adult male donors (Supplementary Data 1). After quality control, we retained data of 49,359 cells in snRNA-seq and 26,714 cells in scATAC-seq, respectively (Supplementary Figs. 1 and 2).

We characterized cell populations with clustering analysis in both snRNA-seq and scATAC-seq datasets. From snRNA-seq[28], we identified eight major cell types based on marker genes and comparison to published single-cell heart atlas data[17] (Fig. 2a, left, Supplementary Fig. 3a), with ~70% of cells from cardiomyocytes (CMs), fibroblasts, and endothelial cells. Clustering based on scATAC-seq data[29] revealed similar cell populations (Fig. 2a, right). We computationally transferred cluster labels from snRNA-seq onto scATAC-seq clusters[28] ("Methods") and unambiguously identified matching cell types (Supplementary Fig. 3b, c). Indeed, expression and chromatin accessibility near marker genes showed high cell-type specificity (Fig. 2b, c). Across the eight clusters, gene scores inferred from scATAC-seq, a metric that summarizes the chromatin accessibility near a gene[29] (Methods), were highly correlated with transcript levels in the matched clusters (Supplementary Fig. 3d). We also found good agreement between cell types identified in our scATAC-seq data and a recent study (Hocker et al.[15]), the only differences in annotation between these two studies was that

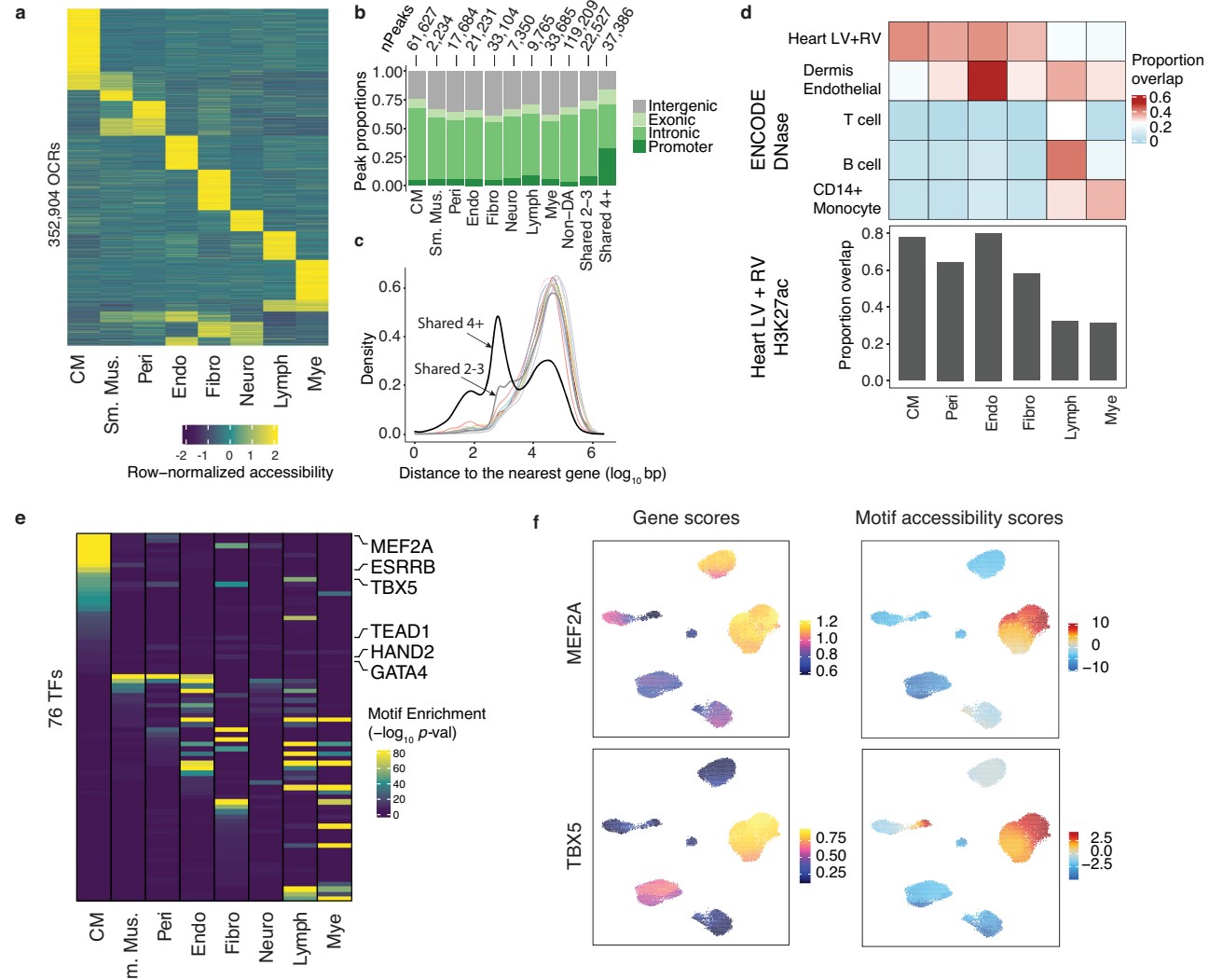

**Fig. 3 | Discovery of OCRs and transcriptional regulators in the human heart.** **a** Row-normalized accessibility of OCRs across all cell types. **b** Number of cell-type-specific and shared OCRs and their genomic distributions. **c** Density plot of the log₁₀ distance to nearest gene for all cell-type-specific and shared OCRs. Colors of the lines for cell-type-specific OCRs follow the same convention as in Fig. 2a. Gray and black lines represent shared 2-3 and shared 4 OCRs. **d** Proportions of cell-type specific OCRs that overlap with DHS (upper panel). Bar graph (lower panel) shows the proportions of cell-type specific OCRs that overlap with H3K27ac regions (LV left ventricle, RV right ventricle). Smooth muscle cells and neuronal cells are not shown due to the small numbers of peaks in these cell types. **e** Enrichment of TF motifs in the OCRs specific to each cell type. Shown are 76 TFs with FDR < 1% from

motif enrichment analysis in at least one cell-type, and correlation between motif enrichment and gene activity >0.5. Enrichment analysis was performed using the peakAnnoEnrichment function in ArchR, which uses the hypergeometric test to assess the enrichment of the number of times a motif overlaps with a given set of peaks, compared to random expectation. After correcting for multiple testing within each cell-type, we used FDR < 1% to ascertain a set of motifs and their enrichment. **f** Gene scores (from ArchR) and motif accessibility scores calculated with chromVar in OCRs for MEF2A (top) and TBX5 (bottom) across all cells. Abbreviations for cell types: CM cardiomyocyte, Sm.Mus. smooth muscle, Peri pericyte, Endo endothelial, Fibro fibroblast, Neuro neuronal, Lymph lymphoid, Mye myeloid.

we did not detect atrial cardiomyocytes, owing to our use of ventricular samples, and that we detected separate pericytes and smooth muscle clusters (Fig. 2a), whereas Hocker et al. annotated a single large cluster as "smooth muscle". These results supported our cell-type assignments in both modalities.

### Analysis of scATAC-seq data identifies cell-type-specific regulatory elements and their regulators

We pooled cells of the same cell type and identified OCRs separately in each cell type. Combining samples of the same cell type (Supplementary Fig. 4a, b), we detected 45,000–150,000 OCRs per cell type (Supplementary Fig. 4c) yielding a union set of 352,904 OCRs. *K*-means clustering of these regions based on their accessibility suggested that most OCRs are active in specific cell types (Fig. 3a). Using differential accessibility (DA) analysis, we identified 173,782 (49%) OCRs with cell-

type-specific accessibility ("Methods"). We divided the remaining 179,122 (51%) OCRs into three categories based on their detection across cell types: shared in 2–3 cell types, shared in ≥4 cell types (denoted as Shared 2-3 and Shared 4), and remaining ones, denoted as "non-DA OCRs", which mostly comprise peaks with low read counts ("Methods"). In agreement with previous observations, shared OCRs were enriched in promoter regions[30] (Fig. 3b, c).

We compared our OCRs to regulatory regions identified in multiple tissues in ENCODE[12]. As expected, a large fraction of OCRs from major heart cell types (e.g., CMs, endothelial, fibroblasts) overlapped with DNase Hypersensitive sites (DHS) from ventricles (Fig. 3d, top, Supplementary Fig. 5a). In contrast, the proportions of OCRs from rare cell types (e.g., myeloid) overlapping with bulk DHS were significantly smaller, suggesting that scATAC-seq is more sensitive and detects more regulatory elements specific to rare cell types compared to bulk

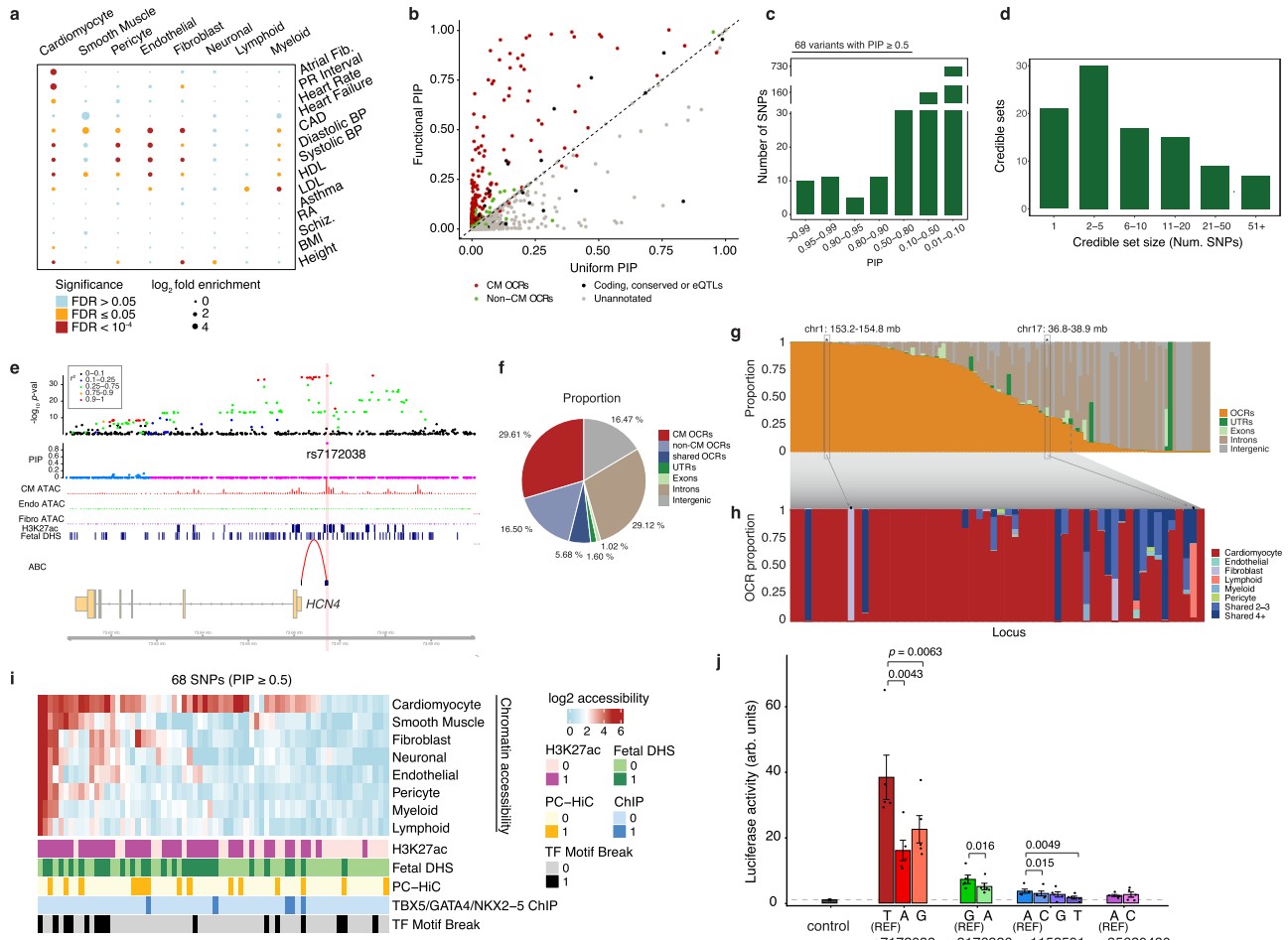

**Fig. 4 | Statistical fine-mapping of loci associated with the AF risk. a** Log$_2$ fold enrichment (from the tool TORUS) of risk variants of various traits in cell-type-specific OCRs. **b** Comparison of AF fine-mapping results under the informative prior using OCRs (Y-axis) vs. the results under the uniform prior (X-axis). Each dot is a SNP, and color represents the annotation of SNPs. Dashed line has a slope of 1. **c** Summary of PIPs of variants. **d** Summary of credible set sizes from fine-mapping of AF. **e** Trackplot at the *HCN4* locus and the fine-mapped variant rs7172038 (PIP = 0.99). The top two tracks represent the -log$_{10}$ p-value of SNPs from AF GWAS (with color representing LD with the lead SNP) and their PIPs from SNP-level fine-mapping. Middle three tracks represent cell-type aggregated ATAC-seq signals (CM: red, endothelial: green; fibroblast: purple), followed by heart H3K27ac and fetal DHS peak calls. The bottom track represents ABC scores from the heart

ventricle. Abbreviations for cell types: CM cardiomyocyte, Endo endothelial, Fibro fibroblast. **f** Proportions of summed PIPs in disjoint functional annotation categories among all the loci. **g** Proportion of summed PIPs in disjoint functional annotation categories at each individual locus. **h** Proportion of summed PIPs into cell type-specific OCRs at each individual locus, for loci with summed PIPs in OCR ≥ 0.25. Highlighted are two loci with high proportions in non-CM cells: fibroblast, lymphoid specific OCRs. **i** Chromatin accessibility and additional functional genomic annotations of all SNPs with PIP ≥ 0.5. **j** Reporter activities in cardiac cells (HL-1) of regions containing selected SNPs, with both reference and alternative alleles. Data are from 5 replicates for each construct. Error bars represent mean +/− SE. P values were calculated using a paired two-sided t test comparing each alternative allele to the reference allele.

DHS (Fig. 3d, top, Supplementary Fig. 4d). This can be seen in several cell-type-specific OCRs near marker genes of rare cell types, which were largely undetected in the pseudo-bulk sample (Supplementary Fig. 6). Additionally, 60–80% of OCRs from major cell types overlapped with H3K27ac regions from LV and RV, suggesting enhancer activities (Fig. 3d, bottom, Supplementary Fig. 5b). Together, these results showed that scATAC-seq identified cell-type-specific regulatory elements. We also compared cell-type-specific OCRs identified in our study to peaks identified by an earlier single-cell study in heart[15] and found that OCRs showed good agreement across studies (Supplementary Fig. 7). Importantly, more than 75% of OCRs detected in our CMs were also detected in ventricular CMs by Hocker et al. (Supplementary Fig. 7).

Chromatin accessibility is largely controlled by lineage-specific transcription factors (TFs)[31]. To identify these TFs, we assessed the enrichment of TF motifs in OCRs specific to each cell type and identified 260 significantly enriched motifs ("Methods"). Because TFs of the same family may share similar motifs, we performed additional

analysis to infer the exact TFs driving the enrichment, assuming that for these TFs, their motif enrichment should correlate with gene expression across cells. To test this, we correlated motif accessibility scores of TFs calculated by chromVar[32] with their accessibility-derived gene scores, a proxy for gene expression[29] (Methods). This analysis yielded 76 TFs with enriched motifs and correlation >0.5 (Fig. 3e and Supplementary Data 2). Many of these TFs are cell type-specific (Fig. 3e) and include known CM regulators, such as TBX5, GATA4, and MEF2A[33] (Fig. 3f). These results provided a compendium of putative transcriptional regulators across major cell types in the human heart.

## Open chromatin regions in CMs are enriched with risk variants of heart diseases and inform statistical fine-mapping

Using our cell-type-resolved OCRs, we assessed the contributions of different cell types to genetics of heart-related traits[34]. Risk variants of AF were almost exclusively enriched (>10-fold) in OCRs from CMs (Fig. 4a). Similar findings were reported in an earlier study[15]. Interestingly, the variants associated with the PR interval showed a similar

enrichment pattern, suggesting a genomic link between PR interval and AF risk for future investigation[35] (Fig. 4a). In contrast, risk variants of cardiovascular traits and diseases, and blood pressure, were enriched across multiple cell types (Fig. 4a). As control, non-cardiovascular traits showed little or no enrichment in heart cell types (Fig. 4a). We also checked the enrichment of genetic risk of AF at open chromatin regions at individual cells, using the method SCAVENGE[36]. This analysis confirms that the vast majority of cells enriched with AF risk are CMs (Supplementary Fig. 8). Together, these results suggest distinct cell type origins of different heart-related traits, highlighting CMs as the main cell type underlying AF.

This observation motivated us to take advantage of the epigenomic data to statistically fine-map causal variants in 122 approximately independent AF-associated loci[37]. We first used TORUS[21] to estimate how putative risk variants are enriched in multiple functional annotations, including protein-coding regions, conserved sequences, and OCRs in CMs (Supplementary Fig. 9a, "Methods")[21]. This information was used to set prior probabilities of variants being causal. We then performed fine-mapping analysis of all AF-associated loci with SuSiE[38]. Compared to fine-mapping that treats all variants equally (uniform prior), this procedure increased the number of high-confidence risk variants. In total, we identified 68 variants whose probabilities of being causal variants, denoted as PIP, are 0.5 or higher, compared with 44 at PIP ≥ 0.5 under the uniform prior (Fig. 4b, c, Supplementary Data 3, and Supplementary Fig. 9b, c). Across 122 loci, our procedure narrowed down putative causal variants to 5 or fewer SNPs in 51 loci (Fig. 4d).

We highlighted the advantage of our functionally informed fine-mapping with some examples. In the locus containing HCN4, several SNPs are in high LD with similar GWAS association statistics (Fig. 4e, top). Our procedure identified a single SNP, rs7172038, as the most likely causal variant (PIP = 0.99) in the locus (Fig. 4e, middle). This SNP is inside a CM-specific OCR, and H3K27ac region in the heart. Interestingly, the Activity-by-contact (ABC) method[39] predicted a loop between the SNP and the HCN4 gene. HCN4 is an ion channel and has been implicated in the genetics of AF[14]. In another example, we nominated a likely causal variant (PIP = 0.96) among several high LD variants in the locus containing GATA4 (Supplementary Fig. 9d), an important TF for AF[40].

A recent study nominated putative causal variants in 12 AF-associated loci by detecting regulatory variants using STARR-seq[14]. We compared these variants with our fine-mapped variants (Supplementary Data 4). Among the 9 loci where fine-mapping identifies one variant at PIP > 0.5, the fine-mapped variants agree with allele-specific variants from STARR-seq in two loci. In the remaining cases, the disagreement is driven by two sources. Most of the allele-specific variants from STARR-seq have much lower GWAS association than fine-mapped variants, suggesting that statistically they are unlikely to be causal variants (see examples in Supplementary Fig. 10). In addition, STARR-seq tested variant functions in vitro, and a few of allelic variants have no regulatory annotations in vivo (Supplementary Data 4). These results added to the emerging picture that in a trait-associated locus, multiple variants may show regulatory effects in vitro[41]. But to identify true causal variants, we believe one should consider both regulatory information and the strength of GWAS evidence.

The fine-mapping results inform how the risk variants are partitioned into various functional categories, such as exons and OCRs in different cell types. The sum of PIPs of all SNPs assigned to a category can be interpreted as the expected number of causal variants in that category. We found that >50% of causal signals are from OCRs and 30% of signals from CM-specific OCRs, highlighting the key role of CMs in AF (Fig. 4f). As expected, exons and UTRs explain only 3% of causal signals.

The same PIP summation approach can also be applied to each locus, with the PIP sum of a functional category, e.g., OCRs or exons, now interpreted as the probability that the causal variant in that locus falls into that category. Using this approach, we estimate that at more than half of all loci, causal variants have >50% probability to localize to OCRs (Fig. 4g). Further partitioning of OCRs into cell-type-level categories (Fig. 3b), we identified 37 loci where the causal signals almost entirely (>90%) come from CM-OCRs (Fig. 4h). With the only exceptions of two loci, CM OCRs explain the causal signals in most of the loci, based on OCR annotations (Fig. 4h and Supplementary Data 5). Together these results highlighted that our approach could identify cell type contexts of individual loci.

## Fine-mapped variants are supported by regulatory annotations and experimental validation

We characterized the regulatory functions of 68 specific variants at PIP ≥ 0.5. The majority (42/68) were located in CM-OCRs (Fig. 4i and Supplementary Data 3). 60% (41/68) of all variants and 86% (36/42) of variants in CM-OCRs overlapped H3K27ac marks in the heart, suggesting enhancer activities (Fig. 4i). 40% of variants (27/68) overlapped with fetal DHS[12], suggesting that these variants may act across fetal and adult stages (Fig. 4i). Additionally, 22% of variants were linked to promoters through chromatin loops in Promoter-capture HiC (PC-HiC) from iPSC-derived CMs[42] (Fig. 4i). Using mouse ChIP-seq datasets of three key cardiac TFs (GATA4, TBX5, NKX2-5)[33], we found that five candidate variants are located in human orthologous regions of TF binding sites, representing 4-fold enrichment over expectation by chance (Supplementary Fig. 9e). We also found that 22% (15/68) SNPs alter binding motifs (Fig. 4i) of one of the 76 TFs we identified as likely transcriptional regulators in heart cell types (Fig. 3e). Together, these results supported regulatory functions of many fine-mapped variants.

We experimentally tested six non-exonic variants with PIP > 0.95 that were located inside CM-OCRs and overlapped with putative enhancers marked by H3K27ac or H3K4me1/3 (Fig. 4j and Supplementary Data 6). Four out of six variant-containing OCRs induced reporter gene expression in mouse cardiac cells (HL-1 cell line)[43,44] (Supplementary Fig. 11a, "Methods"), but not in a fibroblast line (3T3), suggesting cell-type-specific activity of the four OCRs (Supplementary Fig. 11b). Three out of these four variants showed allelic changes of reporter activities in cardiac cells, for at least one alternative allele (Fig. 4j). The most striking effect was observed for rs7172038. Two alternative alleles of this SNP (A and G) strongly reduced activation. The enhancer containing this SNP interacts with the promoter of HCN4 located about 5 kb away, according to Activity-by-Contact (ABC) score[39] (Supplementary Data 3). HCN4 is a well-known AF risk gene and is physiologically implicated in cardiac rhythm control[45]. Consistent with these results, deletion of a syntenic 20 kb region in mice containing this enhancer significantly reduced the expression of HCN4[14]. Notably, in two out of three SNPs with allelic effects, the use of functional information in fine-mapping significantly boosted their PIPs to ≥0.95 (PIP = 0.40 for rs7172038 and 0.41 for rs1152591 under the uniform prior). These experimental results supported regulatory functions of our high confidence variants.

In principle, we expect regulatory variants to affect transcript levels of target genes. Using GTEx eQTL data from the left ventricle (LV), we found that only 31% (21/68) variants are eQTLs (Supplementary Data 7). And only in four cases, the eQTLs showed plausible evidence of colocalization (PP4 > 0.5 using coloc[46]) with the AF risk (Supplementary Data 7). The small overlap of fine-mapped variants with heart eQTLs suggests a limitation of bulk eQTL data to identify regulatory variants, an issue we will address in more detail below.

## A novel computational procedure utilizes fine-mapping results to identify AF risk genes

Despite our fine-mapping efforts, there remained considerable uncertainty of causal variants in most loci (Fig. 4d). Even if the causal variants are known, assigning target genes can be difficult due to long-

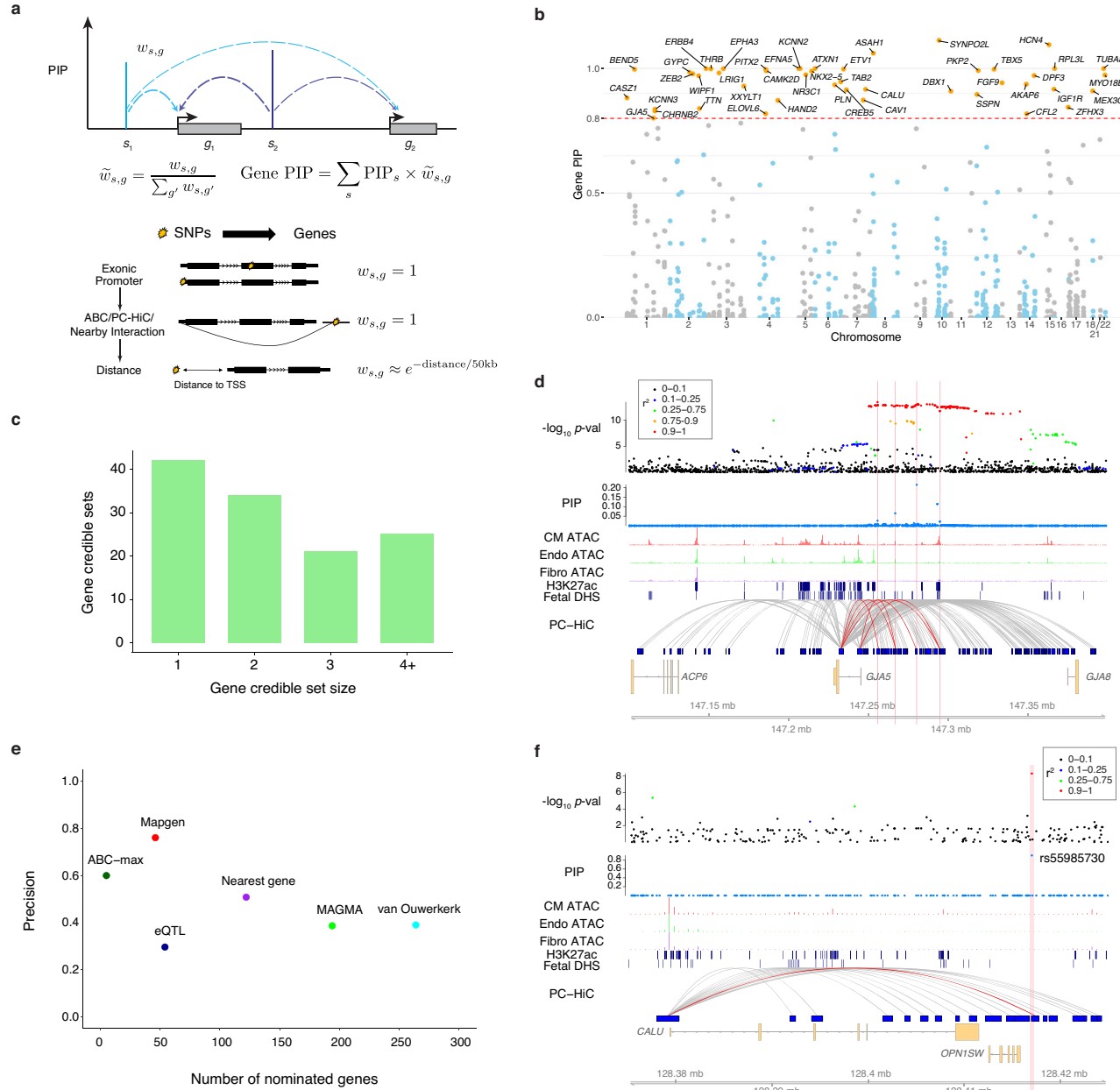

**Fig. 5 | Mapping putative risk genes of AF. a** Schematic demonstrating the calculation of gene-level PIPs. $g_1$ and $g_2$ represent genes, and $s_1$ and $s_2$ represent SNPs. Vertical bars show the PIPs of the SNPs from fine-mapping. Dashed arrows show the linking of SNPs to genes. $w_{s,g}$ represents the weight of a gene ($g$) with respect to a SNP ($s$). The PIP of a gene is the weighted sum of PIPs of SNPs that are linked to that gene. The weight of a SNP-gene pair is set according to the locations of the SNP relative to the gene. See Methods for details. **b** Manhattan plot of gene PIPs. Genes at PIP ≥ 0.8 at labeled. **c** Summary of the sizes of 80% credible gene sets from gene mapping. **d** *GJA5* locus: similar to Fig. 4e, except that the links here represent interactions identified from promoter-capture HiC data in iPSC-derived CMs.

Vertical bars show the locations of the four fine-mapped SNPs supporting *GJA5* as the risk gene in this locus. The red links in the PC-HiC track show interactions linked to these four fine-mapped SNPs. **e** The performance of Mapgen and other methods in nominating AF risk genes. Precision is measured as the proportion of likely risk genes, those annotated with AF-related GO terms, in the set of nominated genes by each method. See Methods for the details of each method. **f** *CALU* locus, tracks shown are similar to **d**. The vertical bar shows the location of the top fine-mapped SNP. Abbreviations for cell types: CM cardiomyocyte, Endo endothelial, Fibro fibroblast.

range regulation of enhancers[47]. We developed a novel procedure, called Mapgen, to address these problems (Fig. 5a, top): (1) For every putative causal SNP, we assign a weight to each nearby gene, considering multiple ways the SNP may affect a gene (see below). The weight of a gene can be viewed as the probability that the SNP affects that gene. (2) For each gene, we then aggregate the causal evidence of all SNPs likely targeting this gene, expressed as the weighted sum of the PIPs of all these SNPs. To ensure that the causal evidence of a variant is not counted multiple times when it targets multiple genes,

we normalize the SNP-to-gene weights in this calculation. The resulting "gene PIP" approximates the probability of a gene being causal ("Methods"). Similar to variant-level fine-mapping, we also define a "credible gene set," the set of genes that capture the causal signal at a locus with high probability ("Methods").

The weights of SNP-gene pairs reflect the strength of biological evidence linking SNPs to genes (Fig. 5a, bottom). For a SNP in an exon or in a regulatory region linked to a particular gene, we assign a weight of 1 to that gene. When a SNP cannot be linked to any gene in these

**Table 1 | Top 15 prioritized genes**

| Gene | Gene PIP | Supporting SNPs | SNP PIP | Link Method | OMIM | CM-specific expression | Known AF risk gene | Reference [PMID] |
|---|---|---|---|---|---|---|---|---|
| *SYNPO2L* | 1.113 | rs60632610 | 0.971 | Exon | | | ✓ | [20215401, 33768119] |
| *HCN4* | 1.095 | rs7172038 | 0.989 | ABC | ✓ | | ✓ | [29987112] |
| *ASAH1* | 1.061 | rs7508 | 1 | Exon | | ✓ | | [32015399] |
| *ATXN1* | 1.000 | rs59430691 | 0.809 | PC-HiC | | | | [21475249, 22306179] |
| *ERBB4* | 1.000 | rs6738011 | 0.12 | Distance | | ✓ | | [19632177] |
| *KCNN2* | 1.000 | rs337705<br>rs337708 | 0.528<br>0.113 | Distance<br>Distance | | | ✓ | [19139040] |
| *RPL3L* | 1.000 | rs140185678 | 1 | Exon | | | | [32870709, 32514796] |
| *TUBA8* | 1.000 | rs464901<br>rs361834 | 0.886<br>0.114 | Nearby OCR<br>Nearby OCR | | | | [31398994] |
| *EPHA3* | 0.999 | rs35124509<br>rs6771054<br>rs2117137 | 0.345<br>0.172<br>0.117 | Exons<br>Distance<br>Distance | | | | [17046737] |
| *THRB* | 0.999 | rs73041705<br>rs73032363<br>rs9841040<br>rs1865712 | 0.177<br>0.139<br>0.130<br>0.119 | Distance<br>Distance<br>Distance<br>Distance | | ✓ | | [28740583] |
| *ETV1* | 0.998 | rs55734480<br>rs12154315<br>rs12112152 | 0.403<br>0.338<br>0.218 | Distance<br>Distance<br>Distance | | | ✓ | [27775552, 29930145] |
| *BEND5* | 0.997 | rs11590635 | 0.973 | Distance | | | | |
| *PITX2* | 0.997 | rs1906615<br>rs7689774 | 0.798<br>0.15 | Distance<br>Distance | | | ✓ | [28217939, 29367545, 32309338] |
| *TBX5* | 0.997 | rs7312625<br>rs883079<br>rs7955405 | 0.511<br>0.194<br>0.126 | Distance<br>Exon<br>PC-HiC | | ✓ | ✓ | [28057264] |
| *PKP2* | 0.992 | rs12809354<br>rs2045172 | 0.771<br>0.211 | PC-HiC<br>PC-HiC | | ✓ | | [28740174] |

In the Supporting SNPs column, only SNPs that contribute a fractional PIP (SNP PIP multiplied by the weight of the SNP to that gene) of 0.1 or more are shown. *EFNA5* (gene PIP = 1) is not included because it does not have any SNPs with fractional PIP ≥ 0.1. Nearby OCR is defined as OCR within 20 kb of active promoter of the gene. Reference column shows the PMIDs of the relevant papers supporting the connections of the genes to AF or heart physiology.

ways, its target genes are assigned using a distance weighted function so that nearby genes receive higher weights ("Methods").

We identified 46 genes with gene PIP ≥ 0.8, and 87 with gene PIP ≥ 0.5 (Fig. 5b, Supplementary Data 8, and Table 1 for top prioritized genes). At each locus, we obtained credible gene sets that captured at least 80% of the causal signal. These credible gene sets contained a single gene in 42 out of 122 blocks, and two genes in 34 blocks (Fig. 5c and Supplementary Data 9). The genes at PIP ≥ 0.8 included many known AF risk genes (Supplementary Data 10), such as TFs involved in cardiac development and atrial rhythm control (e.g., *TBX5*[40] and *PITX2*[48]), ion channels (e.g. *KCNN3*[49]), and genes involved in muscle contraction (e.g., *TTN*).

We note that a key benefit of Mapgen is that even in the absence of high-confidence causal variants, it may still identify putative risk genes. In 14 out of 46 genes at PIP ≥ 0.8, the SNP level PIPs were diffused, i.e., no single SNP reached PIP ≥ 0.5 (Supplementary Data 8). As an example, *GJA5*, a known AF risk gene[50], was supported by seven SNPs (highest PIP = 0.22), five of which were linked to *GJA5* by PC-HiC loops. This led to the gene PIP of 0.80 for *GJA5* (Fig. 5d and Supplementary Data 3). *NKX2-5*, encoding a well-known transcription factor important for heart development[51], was supported by four SNPs (highest PIP = 0.37), all likely targeting *NKX2-5*. This led to a gene level PIP = 0.99 (Supplementary Fig. 12 and Supplementary Data 3). These examples highlighted the advantage of aggregating information from all putative causal variants.

We benchmarked the performance of Mapgen to nominate risk genes against alternative methods. Given the absence of a comprehensive list of known AF risk genes, we used, as a proxy, a set of Gene Ontology (GO) terms previously linked to AF[5]. A gene annotated with

one or more of these terms would be considered as a "true" gene in our evaluation, and otherwise a "false" gene. We considered several methods: nearest gene to GWAS lead SNPs (nearest), Activity-by-contact (ABC) score linking enhancers to promoters (ABC-max), a gene association test method (MAGMA), and heart eQTLs linking variants to genes. Additionally, we included a recent study that nominated risk genes in AF-associated loci based on functional genomic data in heart (denoted as van Ouwerkerk[13]). We found that all these alternative methods, except ABC, have precision below or near 50% (Fig. 5e). ABC score has a precision at 60%, but its sensitivity is low, detecting only a few genes. Mapgen, at the threshold of gene PIP > 0.8, reaches a precision of 76%, while detecting nearly 50 genes. These results thus demonstrated the advantages of Mapgen for risk gene discovery.

We next examined specific loci in detail to gain insights of the weaknesses of existing methods, and how Mapgen addresses them. We focused on the three commonly used methods, nearest gene method, the use of chromatin interaction data, and eQTL. Among the 46 genes at PIP ≥ 0.8, 8 (17%) were not the nearest genes, by distance to TSS, to the top GWAS SNPs. Some of these genes have been implicated in AF and related phenotypes, including *KCNN3*, *TTN* and *HCN4*. Most of the other genes have plausible functions such as *CALU*[52], *SSPN*, and *PKP2* (Supplementary Data 11). Most of the nearest genes in these loci, in contrast, showed limited or no functional relevance (Supplementary Data 11). As an example, in the locus containing *CALU*, the nearest gene of the top SNP, rs55985730 (PIP 0.91) is *OPN1SW*, an opsin gene with function in color vision, but no clear relevance to AF. This SNP is linked to a distal gene *CALU* in PC-HiC data (Fig. 5f), allowing Mapgen to identify *CALU* as the likely risk gene. *CALU* is a calcium-binding protein and involved in alleviation of endoplasmic reticulum (ER) stress in

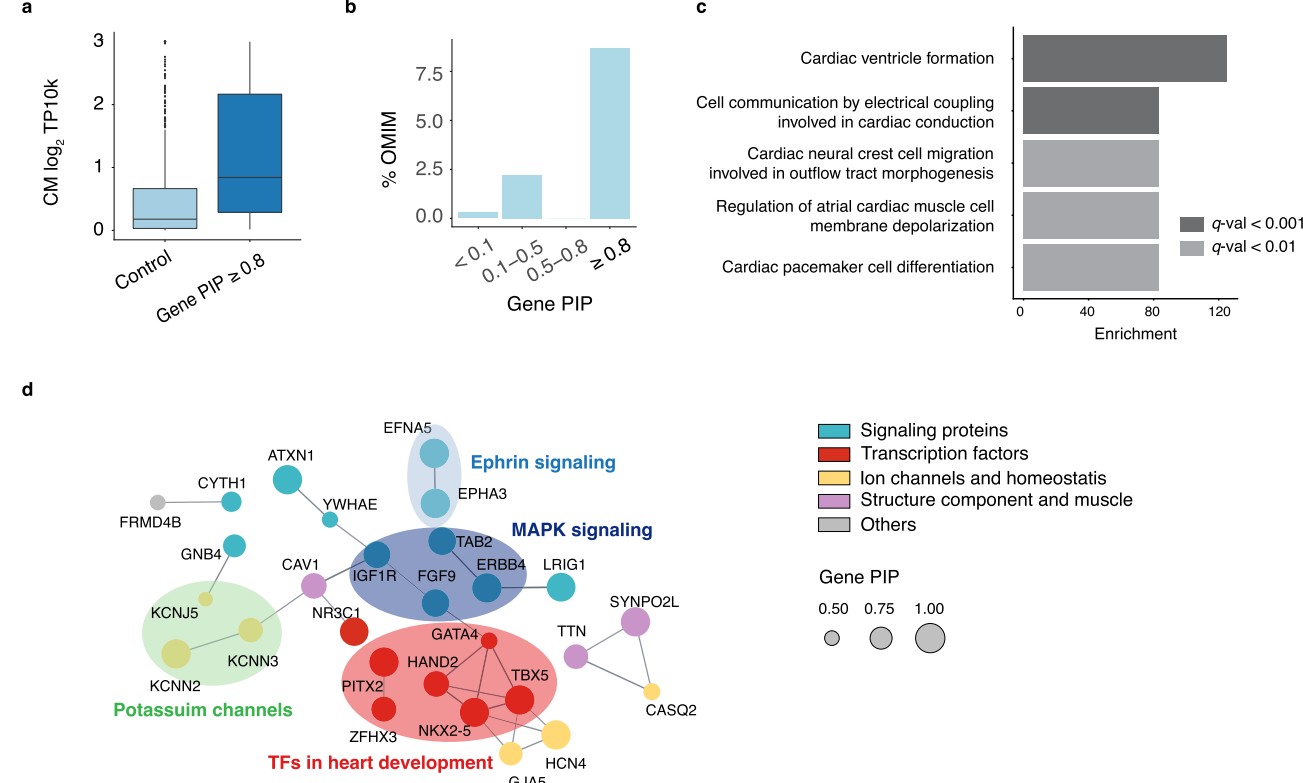

**Fig. 6 | Functional support of putative AF risk genes. a** Log-normalized CM expression of genes at PIP ≥ 0.8 vs. other genes from the AF loci ($n = 580$ for control genes and $n = 42$ for genes at PIP ≥ 0.8). The center line of a box represents the median; the lower and upper hinges of a box correspond to the first and third quartiles; the upper/lower whisker extends from the hinge to the largest/smallest value no further than 1.5× inter-quartile range from the hinge. **b** Percentage of Mendelian disease genes from OMIM in each gene PIP bin. **c** Top 5 Biological Processes (BP) and Molecular Functions (MF) GO terms from gene-set enrichment analysis of the 46 genes with PIP ≥ 0.8. **d** Gene interaction network of candidate AF genes (PIP ≥ 0.5) using STRING. Only genes with interactions are shown. Interactions are defined using a confidence threshold of 0.7 by STRING. Node sizes represent gene PIPs. Colors of genes indicate their shared molecular functions.

cardiomyocytes[53]. ER stress has a critical role in the pathophysiology of AF[54]. These results suggest that by using chromatin loop information, Mapgen is able to identify distal risk genes.

We also considered the use of chromatin conformation in resolving target genes of high PIP SNPs. We found that while chromatin interaction data were useful, as shown in the *CALU* example above, using such information alone may miss many potential risk genes. Among 68 SNPs at PIP ≥ 0.5, only five showed chromatin interactions with promoters based on ABC scores[39], and 18 if we included both ABC and PC-HiC data. Additionally, it is common to observe multiple chromatin loops at a single SNP. Among the 18 SNPs with chromatin interactions, 50% (9/18) contact more than one promoter (Supplementary Data 3), highlighting the uncertainty of target genes from chromatin looping data.

Use of expression QTLs is another common strategy for linking SNPs to genes. However, as reported above, few fine-mapped variants colocalized with eQTLs. Even if a GWAS SNP is also an eQTL, it may not identify the correct target gene. For example, in the *TTN* locus, the top SNP (rs3731746) is an eQTL of *FKBP7*, but the true risk gene is very likely *TTN*[55,56].

Altogether, these results demonstrated the improved ability of Mapgen to nominate plausible candidate genes compared to alternative approaches linking SNPs to genes.

### Putative AF risk genes are supported by multiple lines of evidence

We evaluated our candidate genes using multiple sources of data. Consistent with enrichment of AF variants in CM-OCRs, candidate genes (PIP ≥ 0.8) tended to have higher expression in CMs, compared with other genes in the AF-associated loci (Fig. 6a). Among 46 loci with PIP proportion ≥50% in cardiomyocyte OCR, the likely target genes (gene PIP ≥ 0.5) were highly enriched (nearly tenfold) in cardiomyocyte differentially expressed genes (Supplementary Fig. 13a). Additionally, high PIP genes were enriched in AF-related Mendelian disorders (Supplementary Data 12 and Fig. 6b). We also compared our genes with those prioritized by earlier work that used additional functional data such as AF-related gene ontology and heart gene expression[5,13]. While such functional data was not used in our analysis, the genes at PIP ≥ 0.8 scored on average substantially higher in two earlier studies than low PIP genes (Supplementary Fig. 13b–d), and 32 of them (71%) were supported by at least one study (Supplementary Data 8).

We next assessed the functions of candidate genes using Gene Ontology (GO) and gene networks[57]. GO analysis showed enrichment of Biological Processes related to heart development and cardiac function, and of Molecular Functions such as ion channels, hormone binding and protein tyrosine kinase (Fig. 6c and Supplementary Data 13). For network analysis, we used the STRING gene network built with genes at a relaxed PIP threshold of 0.5 (87 genes) to increase the number of interactions. This analysis highlighted some well-known processes in AF, such as ion channels, and structure components of heart muscle (Fig. 6d). A prominent subnetwork consisted of key TFs, including *GATA4, TBX5, NKX2-5* and *HAND2*, implicated previously in AF genetics and/or heart development[40,58–60] (Fig. 6d). Two other TFs in the network, *PITX2* and *ZFHX3*, are also well-known AF genes[40]. Combined with the fact that putative causal variants were enriched in binding sites of TBX5, NKX2-5 and GATA4 (Fig. 4i and Supplementary Fig. 9e), these results suggested that perturbation of transcriptional

regulatory networks consisting of TFs and their targets, plays a critical role in the genetics of AF. Additionally, the interaction network highlighted signal transduction pathways, including MAPK signaling and Ephrin signaling (Fig. 6d). Both processes are important in heart development[61–64]. Indeed, 19 out of 87 genes at PIP ≥ 0.5 were annotated by the GO term "regulation of intracellular signal transduction" (FDR < 0.02) (Supplementary Data 14).

Finally, we found additional literature support for the candidate genes. 39 out of 46 (85%) genes at PIP ≥ 0.8 have reported roles in cardiac processes and/or diseases from literature (Supplementary Data 10). The top 15 genes with literature support, as well as their supporting SNPs, are shown in Table 1. The majority of these genes have not been established as AF risk genes through functional studies, representing novel yet biologically plausible risk genes.

### Cell-type-specific epigenomes reveal insights to lack of colocalization of GWAS signals to heart eQTLs

While a large fraction of fine-mapped AF SNPs fell inside CM-specific OCRs (Fig. 4f), most of them did not colocalize with heart eQTLs (Supplementary Data 7). This result adds to the growing evidence that eQTLs may explain a relatively small percent of GWAS signals of complex traits[65,66]. It is unclear, however, why this is the case[67]. We took advantage of our cell-type-resolved transcriptomic and epigenomic data to investigate this issue. We hypothesized that the heart eQTLs missed a large fraction of regulatory variants specific to CMs, thus were depleted of AF risk variants. To assess this hypothesis, we focused on 1216 heart (LV) eQTLs from GTEx where the causal variants (known as eQTNs) were fine-mapped with high confidence (PIP ≥ 0.8) (Supplementary Data 15).

Given that the heart eQTL study was performed on bulk tissues, the cell types where these eQTLs act are unknown. We used eQTL information in other tissues to infer whether the eQTLs are likely CM-specific. Our reasoning is that eQTLs that were found across many tissues are likely to be functional in cell types shared across tissues, thus not specific to CMs. We found that the majority of eQTLs were highly shared, i.e., found in >30 tissues in GTEx (Fig. 7a). Less than 10% of heart eQTLs were found in 5 or fewer tissues. These results thus suggest that the detected heart eQTLs are highly biased towards variants with effects in cell types shared across tissues.

This finding thus confirms our hypothesis that the detected heart eQTLs are generally not specific to CMs, even though CMs constitute a relatively large fraction of heart cells (20–30%, Fig. 2a). To understand these results, we divided the heart eQTLs into different categories based on the location of the eQTNs, including exons, UTRs, introns, OCRs in specific cell types, and OCRs shared with varying numbers of cell types. We analyzed the tissue-sharing pattern of each category separately. The overall sharing pattern of all eQTLs would depend on the sharing pattern of each category, and the percent of eQTLs in each category (see "Methods"). This analysis would thus allow us to understand what drives the high degree of observed eQTL sharing across tissues.

As expected, eQTLs falling into OCRs shared in multiple cell types were extensively shared across tissues (Fig. 7b). On the other hand, eQTLs in cell-type-specific OCRs showed variable levels of sharing. Fibroblast-eQTLs (eQTLs in fibroblast-specific OCRs) and myeloid-eQTLs were highly shared (median 31 and 43 tissues, respectively), but most CM-eQTLs were found in <10 tissues (Fig. 7b). We believed this variability reflected different degrees of cell type sharing between the heart and other tissues. To test this, we compared heart eQTLs with those from the brain and whole blood. As expected, heart eQTLs from immune cell OCRs had the highest sharing with whole blood, while eQTLs of all heart cell types have low sharing with the brain (Fig. 7c). Together, these results highlighted considerable variability of tissue sharing patterns of heart eQTLs, depending on their likely cell-type origins.

We next assessed the proportions of heart eQTLs in functional categories, focusing on eQTLs in OCRs, whose cell-type origins could be inferred. A large proportion of those eQTLs were from OCRs shared in multiple cell types, with eQTLs in CM-specific OCRs only explain <10% of heart eQTLs (Fig. 7d). Given that different categories of OCRs have different genome sizes, we compared the proportions of eQTLs in each category with random expectation (Methods). While eQTLs in OCRs from single cell types showed 2–9-fold enrichment, those shared with 4 or more cell types showed 26-fold enrichment (Fig. 7d). Indeed, the enrichment is highly correlated with the number of cell types in which an OCR is detected (Fig. 7e). We thus concluded that discovered eQTLs are biased towards those with broad effects across multiple cell types.

Altogether, our results suggest that eQTLs that are likely CM-specific are under-represented in the data, constituting <10% of all heart eQTLs. Most of the remaining eQTLs have effects across multiple cell types; or have effects in cell types shared across other tissues. As a result, the overall level of tissue sharing of heart eQTLs is very high. Given that AF risk variants are specifically enriched in CM-specific OCRs (Fig. 4a), the depletion of CM-specific eQTLs explains why heart eQTL data fail to explain most GWAS signals.

We reason that this depletion of cell-type-specific regulatory variants in eQTLs can be explained by the nature of bulk eQTL studies. When the effect of an eQTL on a gene is limited to a single cell type, but the gene is expressed in other cell types, the effect of the variant on the bulk gene expression would be diluted, leading to lower power of detecting this eQTL. This argument was supported by the observation that gene expression was less cell-type-specific than accessibility of regulatory elements. In heart eQTLs localized to CM-specific OCRs, the expression of corresponding genes in CMs were only modestly higher than their expression in other cell types (Supplementary Fig. 14a). We performed simulations to investigate the power loss in detecting cell-type-specific eQTLs. When the cell type of interest is 20% of the bulk tissue, we estimate that the power of detecting eQTL specific in this cell type is only about 8-40% of the power of detecting eQTLs with shared effects across cell types (Supplementary Notes and Supplementary Fig. 14b). This analysis demonstrated that the low power of detecting cell-type specific eQTLs is a key limitation of bulk eQTL studies.

## Discussion

While GWAS have been successful in a range of complex traits, the causal variants, their target genes, and their mechanisms in disease-related cell types have been elucidated in few cases[47]. In this work, we established a cell-type-resolved atlas of chromatin accessibility and transcription of the human heart to study the genetics of heart-related traits, focusing on AF[3–5]. We statistically fine-mapped AF-associated loci, and experimentally validated some of the candidate variants. Using a novel computational procedure, we identified 46 high confidence genes, implicating key biological processes, in particular TFs and signaling pathways important for heart development. Motivated by our observation that the putative AF variants often were not colocalized with eQTLs, we investigated how heart eQTLs are shared across tissue types. Our analysis suggests that eQTLs with cell-type-specific effects are under-detected and that this is likely a factor explaining both high tissue-sharing of eQTLs and the lack of eQTLs in GWAS variants.

Compared with several recent studies that aimed to identify risk variants and genes in AF[13,15], our study has a few key advantages. Hocker et al. intersected fine-mapped variants with cell-type-resolved OCRs to nominate putative regulatory variants. Their work and related studies[68] demonstrated the utility of single cell ATAC-seq data for interpretation of non-coding variants from GWAS. Our work extends these studies by using a computational procedure that leverages the strong enrichment of genetic signals in CM-specific OCRs to fine-map causal variants,

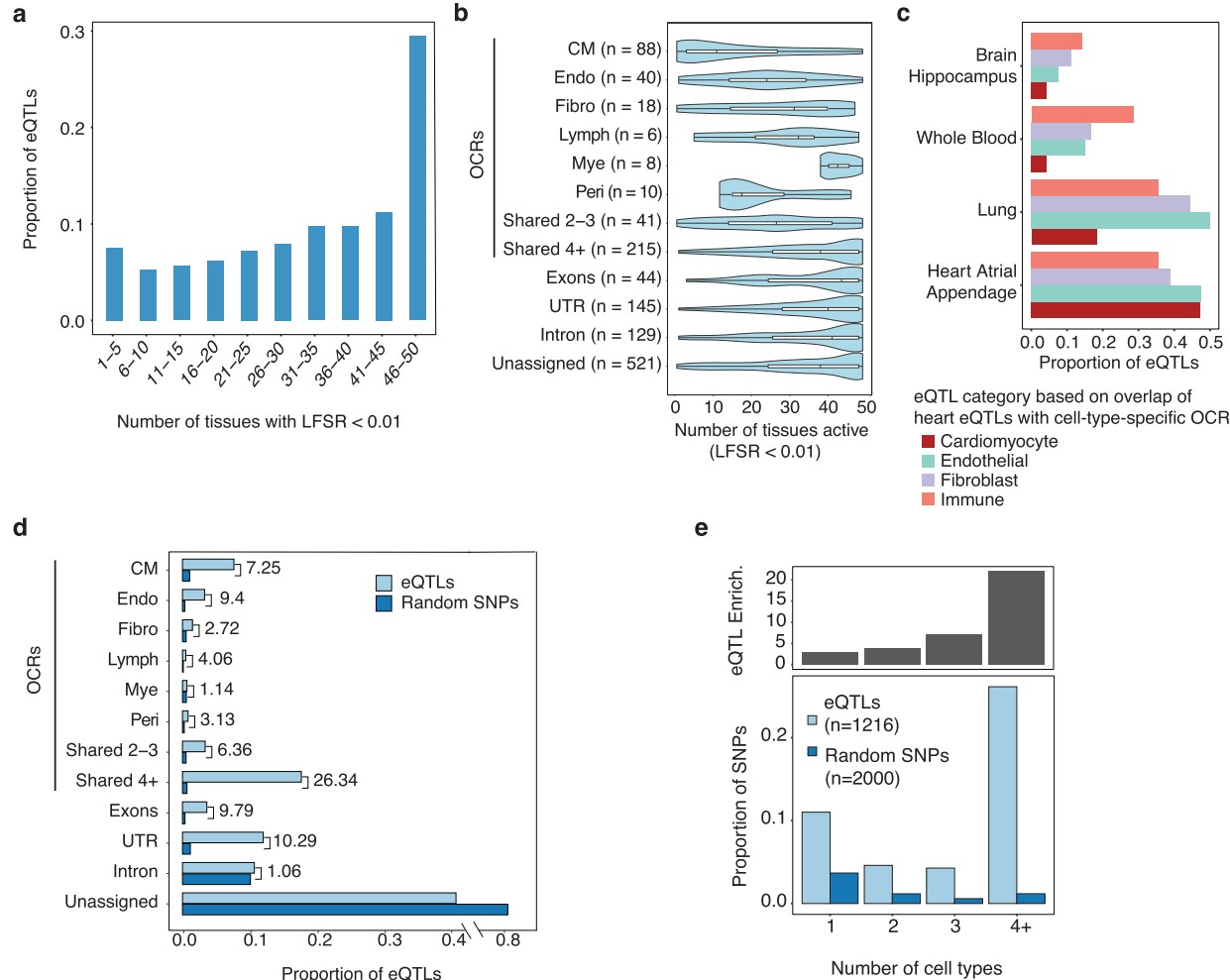

**Fig. 7 | Tissue-sharing patterns of heart (LV) eQTLs from GTEx. a** Number of tissues where LV-eQTLs are detected at local false sign rate (LFSR) < 1%. **b** Violin plot showing the number of tissues in which a specific eQTL is detected. Each row represents a different class of eQTLs, assigned based on their overlap with OCRs categories and other genomic locations. Unassigned: eQTLs that cannot be assigned to any functional class. The center line in the box represents the median; the left and right hinges of a box correspond to the first and third quartiles. The thin black line extending from it represents the smallest and largest values in the data. **c** Proportion of LV-eQTLs located in OCRs of selected cell types (Cardiomyocytes, Endothelial cells, Fibroblast, and Immune cells) that were also detected as eQTLs in a second tissue. **d** Proportion of LV-eQTLs (n = 1216) in each functional class. For

comparison, the proportions of random SNPs in all the classes are also shown. The numbers near the bars represent the fold enrichment in heart eQTLs compared to random SNPs. The numbers of eQTLs in the categories are the same as in (**b**). **e** Enrichment of GTEx heart eQTLs in OCRs vary with the number of cell types where the OCRs are active. Lower panel shows the proportion of eQTLs (light blue) and control SNPs (dark blue, chosen to match eQTLs in LD and MAF) overlapping OCRs. The OCRs are divided into 4 categories, based on the degree of sharing across cell types in heart: 1 = not shared, 4+ = shared in ≥4 cell types. The upper panel shows the enrichment of eQTLs in each OCR class compared to expectation based on control SNPs. Abbreviations for cell types: CM cardiomyocyte, Endo endothelial, Fibro fibroblast, Lymph lymphoid, Mye myeloid, Peri pericyte.

identifying 68 high confidence SNPs at PIP > 0.5, including 42 in CM OCRs, compared to five nominated by Hocker et al. [15] (Fig. 4b). Our gene-mapping procedure effectively leverages fine-mapping results and multiple sources of information linking SNPs to putative targets. This avoids the bias of previous work that only considers one metric, e.g., distance, to link SNPs to genes, and increases the sensitivity of detecting risk genes. As a result, we found high confidence genes (PIP ≥ 0.8) in more than 1/3 of known AF-associated loci.

Our set of 46 candidate genes shed light on the genetics of AF. Earlier linkage studies implicated ion channels and structural proteins, as well as a few TFs[69]. Our results confirmed these earlier findings and showed an even larger role of regulatory genes, including TFs and signaling proteins. In total, we identified 13 TFs with PIP ≥ 0.8 (Supplementary Data 10), and 18 at PIP ≥ 0.5. These included known AF genes, *TBX5* (PIP 0.99), *NKX2-5* (0.99), *PITX2* (0.99), *ZFHX3* (0.85) and *GATA4* (0.57), as well as TFs with roles in heart development such as *HAND2* (0.87), *ZEB2* (0.98), and *PRRX1* (0.72). Our results also

highlighted signal transduction pathways, including MAPK signaling[61], Ephrin signaling[62-64] (Fig. 6d), G-protein coupled receptor signaling[70], Wnt signaling[71] (Supplementary Data 13) and FGF signaling[72,73] (*FGF9*, PIP = 0.94 and *FGF5* PIP = 0.53), all previously implicated in heart development.

Despite the advances described above, our study has a few limitations. Our experimental data were limited to four anatomical locations of the ventricles, while some AF risk variants might act through atrial-specific CMs. However, it is worth noting that a recent study, using scRNA-seq based cellular atlas of the heart including all anatomic locations, found that AF candidate genes were strongly enriched in ventricular CMs[17]. Additionally, our data were from adult hearts, and thus may miss regulatory elements acting transiently during development. Our computational procedure relied on statistical fine-mapping, which may provide mis-calibrated results in practice[74]. To prioritize genes, we used a set of heuristic rules to link variants to genes. This worked reasonably well in our data, but without

comprehensive evaluation it is difficult to know how well it will perform in other settings. This challenge is exacerbated by the lack of a gold standard dataset to evaluate risk genes from GWAS. Lastly, we showed that bulk eQTLs largely missed the effects of our fine-mapped variants and our analysis suggested that one factor may be the reduced power of detecting cell type specific regulatory effects. We cannot, however, exclude other explanations. For example, many variants may act during cardiomyocyte differentiation and therefore would not be detected as eQTLs in adult samples. We believe future eQTL studies across multiple cell types and different developmental stages would help bridge the gap of our understanding.

In conclusion, by combining novel experimental and computational approaches, our study identified a number of risk variants and genes and revealed key insights of the genetics of AF. These data provide a rich resource for future functional studies. Importantly, our analytic framework, including the software for fine-mapping and risk gene identification, may provide a general model for the study of other complex phenotypes.

## Methods
### Data collection
**Nuclei isolation from adult heart tissue.** Heart tissue samples were obtained from National Disease Research Interchange (NDRI) and were stored at −80 °C and kept on dry ice whenever outside of the freezer. University of Chicago IRB reviewed the study and determined that this research does not constitute Human Subjects Research (IRB19-1429, 9/6/2019). We included samples from 4 regions (left and right ventricles, interventricular septum, apex) from 3 male individuals (Supplementary Data 1). Aliquots of each heart sample were prepared from frozen heart tissue using a tissue pulverizer, which was cooled prior to pulverization for 20 minutes over dry ice. Aliquots assayed in this study ranged from 86.7 mg to 141.6 mg. Prior to library preparation, we purified nuclei using fluorescence-activated cell sorting (FACS) to remove debris and minimize contamination from ambient RNA.

Single nuclei isolation was performed on the heart tissue aliquots as described in Litvinukova et al.[17], with some modifications. Single heart aliquots were kept on dry ice until being transferred into a pre-cooled 2 mL dounce homogenizer (Sigma) with 2 ml homogenization buffer (250 mM sucrose, 25 mM KCl, 5 mM MgCl$_2$, 10 mM Tris-HCl, 1 mM dithiothreitol (DTT), 1× protease inhibitor, 0.4 U/μl, RNaseIn, 0.2 U/μl SUPERaseIn, 0.1% Triton X-100 in nuclease-free water). Samples were dounced 25 times with pestle A (loose) and 15 times with pestle B (tight), filtered through a 40-μm cell strainer, and centrifuged (500 × g, 5 min, 4 °C). Supernatant was discarded and the nuclei pellet was suspended in nuclei resuspension buffer (1× PBS, 1% BSA, 0.2 U/μl RnaseIn) and stained with NucBlue Live ReadyProbes Reagents (ThermoFisher). Hoechst-positive nuclei were enriched using fluorescence-activated cell sorting (FACS) on the FACSAria (BD Biosciences), obtaining between 172,500 and 350,000 nuclei while targeting a maximum of 350,000. Nuclei were sorted into 0.75 ml of resuspension buffer. Flow-sorted nuclei were counted in a C-Chip Disposable Hemocytometer, Neubauer Improved (INCYTO) before commencing with library preparation.

**snRNA-seq library preparation and sequencing.** A portion of the sorted nuclei suspension was removed and brought to a concentration of between 700 and 1200 nuclei per microliter. An appropriate number of nuclei were loaded on the Chromium controller (10× genomics) in order to target between 6000–8000 nuclei, according to V3 of the manufacturer's instructions for the Chromium Next GEM Single Cell 3′ Reagent Kits (10X Genomics)[75]. 3′ gene expression libraries were amplified with 15 cycles during sample index PCR. QC was performed on 3′ gene expression cDNA and final libraries using a Qubit Fluorometer (ThermoFisher) and an Agilent 2100 Bioanalyzer (Agilent). Libraries were sequenced on the NovaSeq 6000 (Illumina) or the NextSeq 500 (Illumina) at the University of Chicago's Genomics Facility using paired-end sequencing.

**scATAC-seq library preparation and sequencing.** scATAC-seq libraries were prepared according to v1 of the manufacturer's guidelines for the Chromium Next GEM Single Cell ATAC Reagent Kits (10× Genomics), with the modification that we started from nuclei that were isolated as described above. Between 9300 and 25,000 nuclei were tagmented using Transposition Mix (10× Genomics) at 37 °C for 1 h and loaded on the Chromium controller. We targeted between 6000 and 10,000 nuclei for library preparation. QC was performed on final ATAC-seq libraries using a Qubit Fluorometer and an Agilent 2100 Bioanalyzer. Libraries were sequenced on the NovaSeq6000 or the NextSeq500 at the University of Chicago's Genomics Facility using paired-end sequencing.

### Single-cell genomic data analysis
**snRNA-seq pre-processing.** FastQ files from 12 sequencing experiments were individually processed using an in-house scRNA-seq pipeline dropRunner[76]. Briefly, dropRunner utilizes FastQC[77,78] to obtain quality control metrics followed by fast and efficient alignment to human reference genome hg38 using STARsolo 2.6.1[79] in GeneFull mode with other parameters set to default. STARsolo performs alignment and quantification of gene expression in one package. We quantified expression at the gene level using Gencode v29 gene annotations[80] utilizing both intronic and exonic reads to improve clustering and downstream analyses of the snRNA-seq data. We extracted the raw gene-by-barcode expression matrices output by STARsolo for downstream analyses. We used Seurat 3.2.1[81] in R to analyze the snRNA-seq data. We combined all 12 expression matrices into a single Seurat object together with the corresponding metadata such as donor and anatomical region. To filter low-quality nuclei, we removed barcodes that contained less than 1000 UMI. We also used DoubletFinder 2.0.3[82] with pN = 0.015 and pK = 0.005 to account for doublets, which works by generating in-silico doublets and performs clustering to identify nuclei that fall in the neighborhood of the generated doublets. After quality control, we retained a total of 49,359 nuclei.

**scATAC-seq pre-processing.** FastQ files from 12 sequencing experiments were individually processed using 10× Genomics CellRanger-atac 1.2.0[83]. We used the command cellranger-atac count to align the fastq files to human reference genome hg38, followed by marking and removing duplicate reads, and producing a fragment file containing the mapped location of each unique fragment in each nucleus. We used ArchR 0.9.5[29] to further pre-process the data and perform downstream analyses of the scATAC-seq data. Using ArchR, we converted the fragments file into a tile matrix, which is a bin-by-barcode Tn5 insertion count matrix, using a bin-size of 500 bp. We also generated a gene score count matrix using the "model 42" from ArchR, which aggregates Tn5 insertion signals from the entire gene body, scales signals with bi-directional exponential decays from the TSS (extended upstream by 5 kb) and the transcription termination site, and accounts for neighboring gene boundaries. Gene annotations were obtained from Gencode v29. To filter low quality nuclei, we kept nuclei with at least 5000 unique fragments and a TSS enrichment score of 6. We also used ArchR's doublet removal approach with default parameters, which is based on in-silico doublet generation. We removed nuclei with a doublet enrichment score >1. After quality control, we retained a total of 26,714 nuclei.

**Cell-type identification from snRNA-seq and scATAC-seq.** We performed normalization, dimensionality reduction, and unsupervised clustering on snRNA-seq and scATAC-seq data in order to identify cell-types. For snRNA-seq, we used Seurat's workflow which begins with

converting counts to log2 TP10k values using the NormalizeData function. Next, we found the top 2000 variable genes using FindVariableGenes and used these genes as input features for Principal Component Analysis (PCA). We computed the top 30 principal components (PCs) for each cell and used these for downstream analyses. We observed batch effects due to different donors, and corrected this batch effect. This was done using the RunHarmony function from the Harmony 1.0[84] package with default parameters to regress out the donor variable from the PCs. Next, we used the FindClusters in Seurat with a resolution of 0.2 on the harmony-corrected PCs to define clusters. We also computed the corresponding UMAP to visualize the harmony-corrected PCs in two dimensions. We used previously established cell-type markers in order to map clusters to cell types[17,18].

We performed cell-type mapping for scATAC-seq using the ArchR package. We performed dimensionality reduction on the tile matrix using the top 20,000 bins in terms of count across all cells. We used the function addIterativeLSI with 2 iterations in order to perform latent semantic indexing (LSI) on the scATAC-seq tile matrix and retained the top 50 LSI vectors. Similar to snRNA-seq, we observed batch effects across different donors, and removed this effect using the RunHarmony function. We used addClusters with resolution = 0.2 in order to cluster nuclei based on the harmony-corrected LSI vectors. addUMAP with min.dist = 0.4 was used to compute a two-dimensional representation of the harmony-corrected LSI vectors. We visualized gene activity scores, as defined in ArchR, using the same marker genes as in snRNA-seq to assign clusters to cell types.

### Defining and classifying open chromatin regions

Insertion read counts were aggregated across all cells in each cell-type to form a cell-type pseudo-bulk and peak calling was performed on pseudo-bulk data of each cell-type. Using the function addReproduciblePeakSet in ArchR in conjunction with MACS2[85], a union set of 352,900 peaks were called in total across all cell-types at FDR < 0.1. This set of peaks, called union set, were used for all downstream analyses.

In order to discover cell-type specific regulatory elements, a single-cell insertion count matrix was created using the function addPeakMatrix in ArchR. Cells were grouped into their respective cell-types and differential accessibility (DA) analysis was performed in a one-vs-all fashion, i.e., one cell type vs. all other ones. To perform DA, we used getMarkerFeatures in ArchR with default parameters, which uses the Wilcoxon rank-sum test on the log-normalized insertion count matrix. To control for technical variation, cells from the cell-type group and the group of remaining cell types are matched in terms of TSS enrichment and number of fragments. Using FDR < 10% and log2 fold-change >1, we found about 47% of the union set to be cell-type specific.

For OCRs that were not differentially accessible, we reasoned that these are more likely to be shared. To further stratify these OCRs into different classes, based on sharing among different cell types, we used a simple quantile-based method. First, we aggregated the ATAC-seq counts across all cells within each cell type for each non-DA peak and normalized the counts by the total sum of counts in each cell type. Next, we binarized the peaks within each cell type based on whether they are in the top 25% or not in terms of their normalized counts. In this way, we identify the top 25% accessible peaks in each cell type. Finally, we count how many times a peak is 1, or highly accessible, across cell-types. Through this strategy, we defined three disjoint sets: shared in 2–3 cell types, shared in 4+ cell types and the remaining peaks denoted as "non-DA." The last category corresponds to peaks that are only highly accessible (top 25%) in one cell type but are not found to be differentially accessible based on our criteria above.

### Comparing snRNA-seq and scATAC-seq

We calculated correlation scores of gene expression levels from snRNA-seq and gene activities from scATAC-seq in the following

manner: First, we selected genes that were up-regulated in each cell type according to differential expression analysis of snRNA-seq data. Approximately 3000 genes were identified in this manner. For each gene, the ATAC-seq gene scores and RNA-seq transcript counts, respectively, were aggregated across all cells in each cell-type cluster, followed by a log transformation. We then used the log-transformed pseudo-bulk gene scores and normalized expression levels to calculate Pearson correlation between gene scores and expression across cell types.

### Comparing the cell labels in our study with Litviňuková's et al.

Label transfer was performed using Seurat to compare the labels in our study with that of Litviňuková et al.[17]. The processed scRNA-seq data from Litviňuková et al. were downloaded from https://www.heartcellatlas.org/#DataSources. LoadH5Seurat from Seurat R package was used to convert the h5ad format into a Seraut object. Next, anchors were identified using the FindIntegrationAnchors function and used as reference, which takes the earlier Seurat object as input. Then, we predicted the labels of our cells with the TransferData function, which used the anchors and our scRNA-seq data (also a Seurat object) as inputs and returned the predicted labels for each cell in our dataset. We summarized the number of matched cells with a heatmap, showing the proportion of matched cells in each cluster.

### Comparing the OCRs in our study with Hocker et al.

We compared the OCRs from our dataset with Hocker et al.[15] using peak sets called on individual cell-type clusters. Cell-type level peaks identified by Hocker et al. were obtained from their CARE portal (http://cepigenomics.org/CARE_portal/Cell_Type_Diversity.html). We only included their peak set from ventricular CMs to match our cell types. For each cell type, we computed a simple overlap between peaks from both datasets using GenomicRanges findOverlaps[86] in R. Overlaps for each cell type were represented as Venn diagrams generated with the eulerr R package.

### Identifying putative TFs regulating chromatin accessibility

We used a set of 870 human motif sequence instances from CisBP[87]. These motif annotations were added onto the ArchR object using the addMotifAnnotations function. Next, enrichment analysis was performed for each motif in each cell-type-specific set of peaks, using the peakAnnoEnrichment function in ArchR. The function uses the hypergeometric test to assess the enrichment of the number of times a motif overlaps with a given set of peaks, compared to random expectation. After correcting for multiple testing within each cell-type, we used FDR < 1% to ascertain a set of motifs and their enrichment.

Motif enrichment analysis may find multiple TFs with similar motifs. To reduce the redundancy and identify true TFs that drive gene regulation, we correlated the motif accessibility with gene score activity of each TF, expecting that for true TFs, their expression levels should be positively correlated with accessibility of their motifs across cells. We obtained motif accessibility scores from chromVAR[32] (using the addDeviationsMatrix function in ArchR) for each TF across all cells. We obtained the corresponding TF gene activity scores using the "model 42" by ArchR (see "scATAC-seq pre-processing"). These single-cell-level motif accessibility scores and gene scores, however, are noisy given the sparsity of data at individual cells. We thus used a strategy similar to Cicero[88], by aggregating cells into "metacells" based on similarity using a $k$-nearest neighbor approach. Specifically, we found the $k$ nearest neighbors to each cell using the LSI vectors of the single-cell ATAC-seq data. We only retained sets of metacells that shared a maximum of 25% of constituting cells. Metacells that shared more than 25% of cells were removed at random. Using $k = 100$, we created about 200 non-redundant meta-cells based on these criteria and averaged the motif accessibility scores and gene scores across cells within each meta-cell. We then computed Pearson's correlation between the gene

scores and the motif accessibility scores across meta-cell. We selected all TFs with a Pearson's correlation >0.5.

## SCAVENGE analysis

SCAVENGE[36] was used to calculate for each cell a trait relevance score (TRS) for atrial fibrillation. SCAVENGE was run under default settings, with ATAC-seq peak matrix and fine-mapping results (under the uniform prior) as inputs.

## Testing enrichment of GWAS risk variants in functional annotations

We obtained harmonized GWAS summary statistics for cardiovascular and some non-cardiovascular traits from the IEU OpenGWAS project. We removed SNPs with missing values, SNPs on non-autosomal chromosomes, and indels. Utilizing approximately independent linkage disequilibrium (LD) blocks generated by ldetect[37], we assigned each SNP to one of 1700 LD blocks.

We used TORUS[34] to estimate the genome-wide enrichment of risk variants of GWAS traits in various functional annotations, including cell-type specific OCRs obtained from DA testing, and some generic annotations including coding, retrieved from UCSC Genome Browser database, and conserved sequences from Lindblad-Toh et al. [89]. We ran TORUS on each annotation, one at a time, to get the marginal enrichment reported in Fig. 4a. P values for enrichment were estimated from the 95% confidence intervals returned by TORUS and were adjusted for multiple testing across all traits/cell-types using the Benjamini–Hochberg approach.

## Fine-mapping causal variants in AF-associated loci

We start with a general description of statistical fine-mapping analysis. We assume the trait of interest, $Y$, is related to the genotypes of all variants in a locus by a linear model. Let $X_j$ be the genotype of the $j$-th variant, we have: $Y = \sum_j X_j \beta_j + \epsilon$, where $\beta_j$ is the effect size of the $j$-th variant. Because causal variants in a locus are generally "sparse", it is often assumed that most $\beta_j$'s would be zero. It is easy to see that, under this model, even if a single variant has $\beta_j \neq 0$, other variants in LD with this variant would appear associated with the trait in the standard single-variant association analysis. But in the joint regression model here, once we choose the correct causal variant(s), conditioned on them, the non-causal variants in LD would no longer be associated with the trait. The goal of fine-mapping is then to select as few variants with $\beta_j \neq 0$ as possible to explain all associations in the locus. This "variable selection" step is often accomplished using a Bayesian spike-and-slab prior, which assumes that $\beta_j$ follows a mixture distribution of point mass at 0, and a normal distribution. The mixture proportion of the point mass is typically very large (close to 1), ensuring that at most a few variants would have non-zero effects. Inference of this model is computationally difficult. We used SuSiE in our analysis[38]. SuSiE uses an efficient variational Bayes procedure, and generally outperforms other fine-mapping tools. The main output of SuSiE is the posterior probability that $\beta_j \neq 0$, denoted as PIP.

To run SuSiE on our GWAS summary statistics, we first partitioned the genotype into LD blocks using LDetect[37,38]. Then, we ran the susie_rss() function on each LD block. The input of this function includes GWAS z-scores and the LD matrix for the SNPs in a block. The GWAS summary statistics were available publicly. For the LD matrix, we used out-of-sample genotype information from 1000 Genome Project[90]. We ran SuSiE with $L = 1$, which allows a single causal signal for each LD block and is robust to mismatching LD patterns. We fine-mapped a total of 122 LD blocks in the AF GWAS, each containing at least 1 SNP at genome-wide significance ($P < 5 \times 10^{-8}$).

To incorporate functional information of variants in fine-mapping, we allowed SNPs to have different prior probabilities in SuSiE. Specifically, each SNP has a different prior distribution of $\beta_j$, with the prior probability that $\beta_j \neq 0$, denoted as $\pi_j$, dependent on the functional

information of that SNP. These prior probabilities are estimated using TORUS[21]. Briefly, TORUS assumes that $\pi_j$ is related to the annotations of the SNP through a logistic regression model. These annotations may include, for example, whether a SNP is located in an OCR in CMs, or in an evolutionarily conserved region. The parameter of an annotation in the model encodes the extent to which causal variants are enriched in this annotation. TORUS uses the entire GWAS data of all variants in the genome to estimate these parameters. We included the following annotations in TORUS: CM specific OCRs, CM shared OCRs, CM non-DA OCRs, non-CM OCRs, UCSC conserved, coding, or fine-mapped eQTLs.

## Annotating putative AF causal variants with additional functional data

Fetal DHS and heart H3K27ac data were obtained from ENCODE. PC-HiC interactions were obtained from an earlier study conducted in iPSC derived CMs[42]. Only interactions found in at least two out of three replicates were included. Motif analysis was performed using R motifbreak package[91]. Only "strong" effects on motif scores, according to the package, were considered.

## Assessing regulatory effects of candidate variants by Luciferase assay

Candidate regulatory elements were designed from CM-specific accessibility in hg38 and synthesized by IDT, with either the reference allele or alternative allele(s) (Supplementary Data 6). Sequence was verified and then cloned into the pGL4.23 enhancer luciferase response vector with a minimal promoter. HL-1 cardiomyocytes (received from Claycomb WC[92]) or 3T3 mouse embryonic fibroblasts (ThermoFisher) were co-transfected with luciferase response vector and a pRL control using Lipofectamine 3000, cultured for 48 h after transfection, then lysed and assayed using the Dual-Luciferase Reporter Assay system (Promega). For each construct reporter gene activity was assayed in 5 replicates in HL-1 cells and at least 3 replicates in 3T3 cells.

## Gene mapping procedure with Mapgen

We used the PIPs generated by SuSiE to calculate a gene-level PIP, reflecting the probability that a gene is a risk gene. We assume there is a single causal gene per disease associated locus. Let $Z_g$ be an indicator variable describing whether gene $g$ is causal ($Z_g = 1$) or not ($Z_g = 0$) for the trait. Assuming a single causal SNP per locus, the probability that the gene is causal, which is denoted as "gene PIP", can be then related to the probabilities of SNPs being causal variants:

$$P(Z_g = 1 | D) = \sum_i P(Z_g = 1 | \gamma_i = 1) P(\gamma_i = 1 | D), \quad (1)$$

where $\gamma_i$ is the indicator variable for whether SNP $i$ is causal or not, and $D$ is the GWAS summary statistics. The term $P(Z_g = 1 | \gamma_i = 1)$ is the probability that $g$ is the causal gene if the causal SNP is SNP $i$, and the term $P(\gamma_i = 1 | D)$ is simply the PIP of SNP $i$, or PIP$_i$. So the gene PIP of a gene is a weighted sum of PIPs of all SNPs, weighted by how much that gene is supported by each SNP (see below). Since the PIPs of all SNPs in a block sum to 1, the gene PIP has an upper-bound of 1. In the rare cases where a gene spans two nearby blocks - e.g. when a gene has large introns, the gene PIP may exceed 1, which can be interpreted as the expected number of causal variants targeting the gene $g$.

To calculate the term $P(Z_g = 1 | \gamma_i = 1)$, we consider the location of the SNP $i$ with relation to the gene $g$, as well as functional genomic data linking SNP $i$ with gene $g$. These data were used to assign the weights, denoted as $w_{ig}$, between SNP $i$ and gene $g$, reflecting how likely the SNP $i$ affects gene $g$. For example, if a SNP is inside an exon of a gene, then the SNP-gene will have weight 1. We note that $w_{ig}$ and $P(Z_g = 1 | \gamma_i = 1)$ have different semantics: it is possible that a SNP affects multiple genes

with weights all equal to 1, but there is only a single causal gene supported by any SNP. In other words, for a causal SNP $i$, the conditional probabilities $P(Z_g = 1 | \gamma_i = 1)$ should sum to 1 across all nearby genes $g$. So we need to normalize $w_{ig}$ with:

$$P(\gamma_i = 1) = \frac{w_{ig}}{\sum_g w_{ig}} \qquad (2)$$

To assign the weight terms, $w_{ig}$, we follow these four steps capturing several scenarios where a SNP may affect a gene: (1) If a SNP is in an exon or active promoter (promoter overlapping with OCR) of a gene, we assign the SNP to that gene with weight $w_{ig} = 1$. (2) If a SNP can be linked to a gene's promoter via "enhancer loops", we assign the linked gene with weight $w_{ig} = 1$. Here, "enhancer loops" are defined based on Activity-By-Contact (ABC) scores (constructed from heart ventricle data with ABC scores $\geq 0.015$)[39] and promoter-capture HiC data (from iPSC-CMs[42]. Considering the fact that Hi-C and PC-HiC may miss contacts between close regions due to technical reasons, we also consider a SNP in OCR within 20 kb of an active promoter as an "enhancer loop". (3) If a SNP is in a UTR but not in OCRs, suggesting that the SNP likely regulates the containing gene through RNA processing mechanisms, e.g. RNA stability or alternative polyadenylation, we will assign the SNP to the UTR-containing gene with weight $w_{ig} = 1$. (4) If a SNP is not linked to any gene via the criteria above, we use a distance-based weighting to assign it to all genes within 1 Mb. The weights follow an exponential decay function as below, where $d_{ig}$ is the SNP–gene distance:

$$w_{ig} = e^{-d_{ig}/5 \times 10^4} \qquad (3)$$

The parameter of this weight function, 50 kb, was chosen based on the fact that most enhancers, estimated to be 84% using CRISPR deletion experiments[93], are located within 100 kb of the target promoters. Using a weight of 50 kb here would lead to 87% of weights within 100 kb, with a simple area-under-curve calculation of the weight function above.

At any locus, having PIPs for all the genes in the locus allows us to define the "credible gene set" of the locus, much like the use of the term for SNPs[38]. Simply speaking, the credible set at the 80% level means the minimum set of genes in the locus whose sum of PIPs is greater than or equal to 80%. One complication is that some of the genes in the locus may span another nearby locus, as described above. In this case, while the final reported gene PIP is computed from both loci, we only use the PIP of the gene from the locus of interest to define the credible gene set of that locus.

### Benchmarking performance of different methods for risk gene identification

We compared the accuracy of Mapgen (gene PIP $\geq 0.8$), and several other commonly used methods that nominate risk genes from GWAS (see below). Given that we do not have a gold standard list of known AF genes, we used a set of Gene Ontology (GO) terms that have been associated with AF genetics (using DEPICT method) from an earlier study (Nielsen et al.[5] and Supplementary Table 7). We used FDR < 5% and required three or more genes in a gene set, to select 173 GO terms. We call a candidate gene "plausible", if the gene is annotated with any of those GO terms. Then we compared the precision of the methods, calculated as the number of plausible genes divided by the total number of nominated genes.

We included the following methods in the comparison: (1) Nearest gene to the top GWAS SNP (based on distance to gene TSS). (2) eQTL, linking gene to the top GWAS SNP in each locus using GTEx eQTL from the left ventricle (LV). (3) Activity-by-Contact (ABC) scores, linking promoters with enhancers based on chromatin-looping data. Following Nasser et al.[39], we used the ABC-max approach, linking each top

SNP to the gene with the maximum ABC score. (4) Multi-marker Analysis of GenoMic Annotation (MAGMA), a gene association test method[27]. We ran MAGMA gene analysis and identified genes with Bonferroni adjusted $p$ value < 0.05. (5) In addition to the above methods, we also included the nominated genes (gene score $\geq 11$) from van Ouwerkerk et al.[13].

### Gene interaction network analysis

We used the STRING database (STRING 11.5)[94] to construct gene network. The analysis was done using Cytoscape 3.8.2[95]. The input genes are those at PIP $\geq 0.5$ from our gene-mapping analysis. To create the gene network (Fig. 6d), we use all default settings except that we use the recommended threshold for high-confidence interactions (0.700) for interaction scores. Singletons, i.e., genes not having any interactions with other ones, were not shown from the output network. We also used STRING to run functional enrichment analysis based on sources including Gene Ontology[96,97], Reactome Pathways[98], and KEGG[99].

### eQTL tissue sharing analysis

We started with the rationale of our eQTL tissue sharing analysis. For simplicity, consider eQTLs found in one tissue (heart in our case), and we study the sharing of these eQTLs in a second tissue. Let $p$ denote the probability of eQTLs in the first tissue being shared in the second tissue. Assuming we have several functional categories of eQTLs, e.g. regulatory elements specific in a cell type, or shared across cell types, we can then break down $p$ into several categories with the simple relation:

$$p = \sum_c p_c w_c, \qquad (4)$$

where $c$ denotes a category, $p_c$ is the probability of tissue sharing in eQTLs from category $c$, and $w_c$ is the proportion of eQTLs in category $c$. We hypothesize that different eQTLs categories have distinct molecular mechanisms of modulating transcript levels, and thus different tissue sharing patterns. This simple analysis thus suggests that both $w_c$ and $p_c$ are important for our understanding of tissue sharing. For instance, some categories may have a highly tissue-specific pattern (low $p_c$), but may constitute a small proportion of all eQTLs (low $w_c$), thus these categories would have limited contribution to the overall level of tissue sharing among eQTLs.

**Summary statistics of GTEx heart eQTLs.** Summary statistics of eQTLs from the left ventricle were obtained from the GTEX v8 release[100]. We also obtained fine-mapping results using DAP-G[21]. The variants with PIP > 0.8 were kept for downstream analyses. We refer to these putative causal variants as eQTLs henceforth. The total number of eQTL-gene pairs that passed the threshold is 1216. Tissue sharing data on the same eQTLs were also obtained from GTEx[100]. These data provide information of whether these heart eQTLs are also associated with gene expression in the other tissues in GTEx.

**Defining functional categories of heart eQTLs.** eQTLs were intersected with genomic features. To obtain a set of disjoint genomic features, we used a combination of the union peak set and generic annotations. For generic annotations, the longest transcript was chosen for each gene body, and its corresponding exons, UTRs, and introns were obtained for all protein coding genes. We partitioned the union peak set into cell-type-specific categories based on the differential accessibility (DA) analysis, as well as the shared categories defined using the quantile approach, as described earlier. We note that DA analysis does not guarantee disjoint sets of features. Indeed, we find that cell types such as lymphoid and myeloid share about 6% of their DA peaks, while CMs share at most 1% with the other cell-types. To

**Article**

make these cell-type DA sets disjoint, we moved any DA peaks that occurred in multiple cell types from DA analysis, to the "Shared 2-3" and "Shared 4+" categories (see "Defining and classifying OCRs") depending on the number of cell types in which it occurred. A small percentage of peaks (<1%) were affected by this step. The eQTLs in OCRs that overlap with exons or UTRs, or eQTLs in non-DA OCRs, are ambiguous to assign, so they were filtered from our analysis. The eQTLs in intronic OCRs were assigned based on the OCR categories. Those eQTLs not intersecting with any functional category were designated in an "unassigned" category.

**Estimating extent of tissue sharing in different categories of heart eQTLs.** GTEx has performed eQTL mapping jointly across all tissues. Using these results, we call a SNP an eQTL in a given tissue, if it passes the local false sign rate (LFSR) threshold of 1%. For any eQTL, we can thus determine the number of tissues where it is active.

**Estimating eQTL enrichment in functional categories.** All the fine-mapped heart eQTLs are assigned to our set of categories. The proportion of eQTLs in each category is then compared with the expected proportion by chance to obtain enrichment reported in Fig. 7d, e. We used SNPsnap[101] to create a set of random control SNPs that match our eQTLs in LD and minor allele frequency. The LD data are obtained from the European population genotypes from 1000 Genomes. We generated 1000 random SNPs which is roughly how many high-confidence eQTLs were used. The proportion of random SNPs in each category is then used as our estimated proportion by chance.

### Reporting summary
Further information on research design is available in the Nature Portfolio Reporting Summary linked to this article.

## Data availability
The snRNA-seq and scATAC-seq data generated in this study have been deposited in the GEO repository under accession code GSE224997.

## Code availability
Mapgen R package is available from https://github.com/xinhe-lab/mapgen. Codes for data processing and analyses are available at https://github.com/xinhe-lab/aFib_heart_atlas_mapgen_paper[102].

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

## Acknowledgements

This work was funded by National Institutes of Health (NIH) grants, R01MH110531 and R01HG010773 (to X.H.), R01HL163523 (to X.H., S.P., and I.P.M.), and R21 AI144417-02 (to A.B). This project has been made possible in part by grant number CZF2019-002431 from the Chan Zuckerberg Initiative DAF, an advised fund of Silicon Valley Community Foundation. This work was completed in part with resources provided by the University of Chicago Research Computing Center. We thank Xuanyao Liu for helpful comments on the manuscript.

## Author contributions

X.H. and S.P. conceived the idea and supervised this project. A.S. and K.L. developed the method, implemented the software, and performed data analyses. M.W., L.S., and H.E. conducted the experiments. X.S. and C.T. performed data analyses. M.W., L.S., and X.S. contributed equally. S.P., I.P.M., and A.B. supervised experiments. A.S., K.L., X.H., and S.P. wrote the manuscript.

## Competing interests

The authors declare no competing interests.
