## [Peer Review File · Nature Communications]

Single-cell genomics improves the discovery of risk variants and genes of Atrial FibrillationEditorial Note: Parts of this Peer Review File have been redacted as indicated to remove third-party material where no permission to publish could be obtained.

Reviewers' comments:

Reviewer #1 (Remarks to the Author):

Single-cell genomics improves the discovery of risk variants and genes of cardiac traits" by Alan Selewa et al.

Selewa et al have performed single nucleus RNA-seq and single cell ATAC-seq to obtain cell-type specific transcriptomes and accessible chromatin sites (OCR, open chromatin regions) of the major cell types of the human heart ventricles. They found that the SNPs associated with atrial fibrillation and with PR interval were highly enriched in cardiomyocyte OCRs. They then used the cardiomyocyte-specific OCR data and additional features to statistically fine map causal variants in 122 AF-associated loci. In total 54 variants with relatively high probability of being causal were identified using this computational approach. The functionality of the fine-mapped variants was supported by literature and additional correlating data, such as co-localization with H3K27ac signal and promoter capture HiC data. They then used the fine mapped SNPs to identify AF risk genes using a novel computational approach. 45 (PIP \geq 0.8) and 88 genes (PIP \geq 0.8) were identified, and validated through literature and computational network analysis. Finally, they provide insight into the extensive tissue-sharing of bulk eQTL, the limitations of current eQTL studies and highlights the need of other strategies such as single-cell eQTL mapping.

1) In general, the single cell ATACseq data of human heart is a potentially valuable resource. snRNA-seq of human heart tissue is readily available already. Previous published studies have provided candidate functional AF-associated SNPs, experimental validation of those SNPs, and candidate AF associated regulatory SNP target genes. The current study also provides such candidate SNPs and genes, but does not meaningfully assess the performance of their identification methods compared to these previous efforts, and does not provide experimental evidence that indeed functional SNPs and AF-associated genes were identified. Therefore, the selected SNPs and genes largely remain theoretical candidates.

2) the correlation between cell type-specific OCR and expression (extended figure 3c) is much higher than the correlation normally found. As accessibility is only one aspect of transcriptional activity, and quantitatively does not strongly correlate with expression level, the observed high correlation is remarkable. Please specify how the correlation was made, which parameters were used, whether quantitative data was used (level of expression, level of accessibility signal from ATACseq).

3) The fine mapping method of AF SNPs is not entirely clear to me. The authors focus on almost 900 highly AF associated SNPs. Previous work has indicated that also the large number of sub-threshold SNPs within associated variant regions and in LD with the highly significantly associated SNPs can be functional/affect regulatory element function (Wang et al, eLife 5, 2016). As these were not included in this analysis, a large number of causal SNPs are likely to remain undetected. Indeed, only 54 SNPs were marked to be probably functional in this study. Furthermore, the fine-mapped SNPs should be experimentally validated. Also, the results should be compared with those of previously published efforts to identify causal AF associated SNPs such as van Ouwkerk et al., Circ Res 2020 using STARR-seq of a number of AF associated loci.

4) The fine mapping of SNPs seems to rely on the OCRs identified in the scATACseq data from the clustered ventricular cardiomyocytes. Something very similar has been done already in van Ouwkerk, Nature Communications 2019. In this study, ATACseq data from purified human atrial cardiomyocytes was used to identify AF associated SNPs (including the sub-threshold ones) that co-localize with atrial cardiomyocyte-specific OCRs and with human cardiac enhancers. The authors should compare their results with the mapping in this study, provide arguments how their approach is complementary, how it performs compared to this and other studies mapping AF SNPs, and provide experimental evidence that their fine mapping indeed enriches for functional SNPs.

5) It was not clear to me which SNPs were used to identify the AF risk genes. Did the authors use the small set (of 54) fine-mapped SNPs? If so, this analysis could be biased by the stringent but un-validated selection method, leading to under-reporting of candidate genes. Moreover, while the authors compared their results with those of Nielsen et al and van Ouwerkerk et al., they do not draw any conclusions from this comparison, and do not provide insight into the relative performance of the three methods to identify AF-associated variant target genes. Again, experimental validation of their candidates is lacking.

6) the section on eQTLs (lane 333 and further) was difficult to follow for me. Please consider clearer description and/or a figure explaining the topic using an example.

Reviewer #2 (Remarks to the Author):

The manuscript by Selwa, et al describes the use of single-cell chromatin accessibility data in cardiac tissue to improve on the fine-mapping of putative casual variants and their target genes at GWAS loci for atrial fibrillation (AF). They also investigate the challenges of detecting of eQTL in bulk tissue studies. The primary contributions are the novel adaption of Bayesian fine mapping to include a prior based on information from the single-cell chromatin data, and the novel method to determine target genes for fine-mapped variants. Though there is a focus on AF, it isn't as clear whether specific proposed casual variants and/or target genes represent a significant contribution to understanding this disease. Validation experiments do confirm the effects of proposed genetic variants on regulatory activity. The results presented provide evidence that these proposed new methods can help in better determining how genetic variation may be contributing to phenotypes through altered gene regulation.

My primary concern and difficulty in evaluating the results presented are that while these new methods demonstrate characteristics desired in improved fine mapping (i.e. confidence measures indicating a greater degree of accuracy, fewer variants in credible sets) and in target gene prediction (distributed evidence, evidence of function in disease-related processes), there is no real empirical evidence showing that the predictions are accurate or at least more accurate than other methods. Understandably, performing functional experiments on predicted variants/genes to more definitively show mechanism are difficult, but either arguments for why these methods are more accurate than something else need to be made clearer, or experiments (could be analytical) need to be designed to provide more evidence. Some more specific comments follow, not in order of importance.

1) There are at least two other single-cell chromatin studies in heart tissue. One is the recently published human atlas (<https://pubmed.ncbi.nlm.nih.gov/34774128/>) and one more focused on cardiac cells (<https://pubmed.ncbi.nlm.nih.gov/33990324/>) that also investigated AF. How do the cell type clusters in the current study compare to those? Do you find essentially the same clusters, or are there any discrepancies? Is there significant agreement in which regions are considered accessible in like cell groups? These are focused on establishing how consistent single-cell ATAC is across multiple studies, and thus generalization of results presented here, as well as supporting the accuracy of the cell clusters.

2) Similar to the above, OCRs are compared with bulk tissue DHS, but why not also compare with the existing single-cell studies?

3) The previous study on cardiac cells (<https://pubmed.ncbi.nlm.nih.gov/33990324/>) also reported the enrichment of AF risk variants in cardiomyocytes. This should at least be acknowledged.

4) PR interval first mentioned on line 164, but is not defined and it's not clear why this phenotype is even mentioned since it is not really discussed in any detail.

5) Using a prior to favor variants in certain regions is shown to increase the number of high confident variants (PIP ≥ 0.5). But is this simply a result of having a prior and not necessarily the prior providing useful additional information? What if you randomly assigned priors? Or assigned them based on distance to the reported lead SNP? It would be good to better show that your specific priors are contributing in a unique way, especially due OCR information.

6) You state that your fine-mapping method can help determine the cell types a variant may be most relevant in. Can you use your single-cell RNA-seq data to further support the mappings of variants to cell types? You give a couple of examples (IKZF3, TNFSF13) that based on their known functions agree with the cell type mapping, but can this be done in a more systematic way? Are the relevant OCR annotations for these variants enriched, say, around genes that are also cell-specific or differentially expressed in that cell type?

7) HL-1 cells should be defined as a cardiac muscle cell line to make the relevance obvious. Figures for reporter assays should state how many replicates were used in creating the plots/error bars.

8) You state that the lack of overlap between your fine-mapped variants and heart eQTL in bulk tissue suggests a limitation in of the eQTL – but why is this necessarily true and not that your fine mapping is wrong?

9) You present your Mapgen method that you propose is more accurate at assigning target genes. But, you don't really present much evidence that this is more accurate than the simple method of assigning the closest gene. In fact, 39 of 45 of your predictions were the closest gene. Your main evidence for the other 6 is that 5 have annotations that show their involvement in the heart. But for these 6, are the closest genes also involved in heart function? You show one has PC-HiC evidence, which is really the only one that shows good evidence of accuracy. For rs1152591, why is uncertainty better? This is only true if all four potential genes are involved. The arguments for not using chromatin looping or eQTL are rather weak. While these may not be comprehensive, it's not clear how detrimental it would be to try and incorporate that information.

10) There is a repeated claim that sharing of eQTLs across tissues is "surprising" or "puzzling", which I'm not sure reflects a general consensus in the community. Admittedly, the current studies in tissues do find a certain amount of overlap across tissues, but is the amount really surprising? What was the prior expectation? The issue of tissue heterogeneity is well-known, and in the latest GTEx analysis, they specifically attempted to investigate the influence of this. In their study, they found "the pairwise relatedness of GTEx tissues derived from their cell type composition is highly correlated with tissue sharing of regulatory variants (cis-eQTL versus cell type composition Rand index = 0.92)". This suggests that common eQTL in tissues are due to common eQTL in underlying cell types, thus this idea that there is a lack of awareness that performing eQTL on tissues misses underlying eQTL in specific cell population seems wrong. The analysis does nicely present some observations about the effect of performing tissue eQTL, but I think the significance of these observations is a bit over-stated.

11) In Figures where abbreviations for cell-types are used as labels, these should be defined in the figure legends.

Reviewer #3 (Remarks to the Author):

In this manuscript, Selewa et al. develop an integrative framework by leveraging single-cell genomic and genome-wide association study (GWAS) data to prioritize risk variants and their potential target genes. The authors first generated single nuclear RNA-seq (snRNA-seq) and single-cell ATAC (scATAC-seq) data of the human heart and identify cell type-specific cis-regulatory elements (CREs). Open chromatin regions in cardiomyocytes are enriched with risk variants for atrial fibrillation (AF). By considering the colocalization of variants and CREs, the authors perform functional fine-mapping to

identify likely causal variants through an existing method named SuSiE. A new computational procedure named Mapgen was developed for nominating the target genes, which can calculate a gene-level probability reflecting how likely a gene may be implicated in underlying the GWAS signals. Finally, the resulting fine-mapped SNPs and their nominated target genes were analyzed and validated by analyses, including genomic interaction, gene-gene interaction network, and eQTL data.

With the increasing availability of functional genomic data at single-cell resolution, integrative analyses of single-cell genomic data, particularly epigenomic data, with GWAS is important for interpreting the function of genetic variants and defining the etiology of disease. This study profiles both single-cell transcriptomic and epigenomic data in the adult human heart, which can be a useful resource. The new computational procedure named Mapgen has the potential to be scalable to other studies. Overall this is a valuable paper that should be published, but a few concerns need to be addressed before publication, as I note below.

Major Comments:

Data quality and accessibility:

1. The cell type specific marker genes from the snRNA-seq data appear lowly expressed as shown in Fig 2c. Can more validation of the cell type specific clusters be performed to bolster this further?
2. It is surprising that most open chromatin regions (OCRs) with cell-type-specific accessibility are enriched in intronic regions (Fig. 3b), although a high enrichment in the non-coding regions makes sense. Is this observation consistent with prior studies? If not, could the author discuss why this may be the case.
3. Given that the numbers of cell type-specific peaks across different cell types are not the same, it does not appear to be valid to just show the proportions of overlap with bulk level data (Fig. 3d). The odd ratios are useful to correct differences in peak numbers. However, it would be helpful to show how many informative peaks are exclusively identified by single-cell data.
4. The scRNA-seq and scATAC-seq generated by this study is a valuable resource. The processed data should be made publicly available and ideally a reviewer code would have been provided, as currently it is only listed as something that "will be deposited to the Gene Expression Omnibus."

Functional fine mapping and interpretation:

1. Functionally informed fine-mapping is performed by using an existing method called SuSiE. The use of this method should be discussed explicitly in the results section rather than only mentioning this in the methods and methodologic credit should be provided.
2. The author use fine mapping by integrating OCRs. Assigning priors to variants according to OCRs for fine mapping can increase the interpretability (Fig. 4b). It is important to show how many loci will be missed by this approach compared to traditional fine mapping? The author can illustrate chromatin accessibility and additional functional genomic annotations that are similar to Fig. 4h but for SNPs called from traditional fine mapping?
3. One recent method (10.1101/2022.01.23.477426) enables genetic variant relevant cell type/state discovery at single-cell resolution by using scATAC-seq. Can the authors apply this approach to their data and compare performance to the mapping methods they have employed?

Target gene identification:

1. Only 6 of 45 genes at $PIP \geq 0.8$ are not the closest genes to the likely causal SNP. Prior studies have shown that nearly half of the target genes are not the closest genes in some cases. Can the authors address these inconsistencies? Ideally, the SNP-target gene pairs should be benchmarked by using Hi-C data or other experimental evidence.

eQTL analyses:

1. The eQTL analyses are interesting and suggest the importance of functional interpretation of variants with single-cell data. However, some statements made appear to be overstated. It is questionable whether bulk-level data is truly under powered for the discovery of cell-type-specific

eQTLs. Prior studies (10.1016/j.ajhg.2018.04.011) have shown that about 12% of 23,000 eQTLs in blood are cell-type-specific. This is also evident in the analysis in Fig. 6b, where CM-related eQTLs showed strong specific activity only in several tissues.

Minor Comments:

1. In Fig. 2a, the data from different donors and from different sources cannot be distinguished in the stacked bar plots. In Line 118 the authors suggest a comparison with another datasets (<https://www.nature.com/articles/s41586-020-2797-4>), but how/where this is shown is not clear.
2. In Extended Data Fig. 4a,b, the lower boundary is 0.8 and the indicated color is white while there is no white color in the heatmap.
3. Lines 143-145. This observation should be expected as the cell-type-specific peaks are used and is unlikely to be due to the fact that myeloid cells are rare.

We thank the reviewers for their thoughtful comments and their suggestions. In response to these comments, we substantially revised our manuscript. We believe that these changes addressed the comments and greatly improved the manuscript. We summarized the main changes below:

- Discussion of related work that aimed to identify risk variants and genes underlying genetics of atrial fibrillation (AF). We added a new paragraph in Introduction, and updated part of Discussion.
- In the method overview section (the first section of Results), we provided some background on the statistical fine-mapping technique. We also added a new section in Methods with details of how fine-mapping was performed in our study.
- Comparison of our data with published single-cell dataset in the heart, validating our cell type assignments.
- We performed a benchmark study of our gene discovery tool, Mapgen. The results show that Mapgen performs much better than existing tools.
- We largely revised the introductory part of the eQTL section (the last section of Results). Our focus now is to understand why fine-mapped AF risk variants are often not heart eQTLs.
- We recognized a minor issue in the ATAC-seq data we used for statistical fine-mapping in our submitted manuscript. We have updated the fine-mapping and gene discovery results. The impact on the results are modest.

Some reviewers questioned the rigor of our method for variant prioritization and candidate gene identification. Given that this is a shared concern, we addressed it here before responding to each reviewer below. Our main analysis was based upon statistical fine-mapping. Fine-mapping has been widely used as a way of identifying causal variants from GWAS. The advantages of fine-mapping over heuristic approaches to nominate risk variants have been well-documented in literature, e.g. Schaid et al, “From genome-wide associations to candidate causal variants by statistical fine-mapping”, *Nature Reviews Genetics*, 2018. Recent developments in this field also allow researchers to incorporate functional information of variants to improve the accuracy of fine-mapping. Because of these advantages, fine-mapping has been increasingly used as a rigorous way of risk variant discovery in GWAS. We provide a few studies from literature here as examples:

- Mahajan et al, Fine-mapping type 2 diabetes loci to single-variant resolution using high-density imputation and islet-specific epigenome maps, *Nature Genetics*, 2018
- Ulirsch et al, Interrogation of human hematopoiesis at single-cell and single-variant resolution, *Nature Genetic*, 2019
- Weissbrod et al, Functionally informed fine-mapping and polygenic localization of complex trait heritability, *Nature Genetics*, 2020
- Fachal et al, Fine-mapping of 150 breast cancer risk regions identifies 191 likely target genes, *Nature Genetics*, 2020

- Zhang et al, Allele-specific open chromatin in human iPSC neurons elucidates functional disease variants, Science, 2020 (from our own work)
- Schwartzenuber et al, Genome-wide meta-analysis, fine-mapping and integrative prioritization implicate new Alzheimer's disease risk genes, Nature Genetics, 2021

Our variant fine-mapping procedure largely followed the literature. Our specific strategy of incorporating functional information was based on TORUS (Wen et al, AJHG, 2016), a method that has been used in multiple studies, including our own (Zhang et al, Science, 2020, cited above), and the GTEx consortium (GTEx, Science, 2020). So we believe that our variant mapping effort was rigorous and provided high-confidence variants. With regard to gene discovery, our novel method (Mapgen) addresses some fundamental limitations of the current methods of risk gene discovery. The conceptual problems of existing methods have been discussed in our method overview section. We have now added a benchmark study showing that Mapgen performs better than existing methods. These results were added in the section “A novel computational procedure utilizes fine-mapping results to identify AF risk genes”. We copied the text here.

We benchmarked the performance of Mapgen against alternative methods to nominate risk genes. Given the absence of a comprehensive list of known AF risk genes, we used a set of Gene Ontology (GO) terms previously linked to AF (Nielsen et al, Nature Genetics, 2017) as a proxy. A gene annotated with one or more of these terms would be considered as a “true” gene in our evaluation, and otherwise a “false” gene. We considered several methods: nearest gene to GWAS lead SNPs (nearest), Activity-by-contact (ABC) score linking enhancers to promoters (ABC-max), a gene association test method (MAGMA), and heart eQTLs linking variants to genes. Additionally, we included a recent study that nominated risk genes in AF-associated loci based on functional genomic data in heart (denoted as van Ouwerkerk, van Ouwerkerk et al. Identification of atrial fibrillation associated genes and functional non-coding variants. Nat Communications, 2019). We found that all these alternative methods, except ABC, have precision below or near 50% (Fig. 5e, copied as Figure 1 below). ABC score has a precision at 60%, but its sensitivity is very low, detecting only a few genes. Mapgen, at the threshold of gene PIP ≥ 0.8 , reaches a precision of 76%, while detecting 46 genes. These results thus demonstrated the advantages of Mapgen for risk gene discovery.

Figure 1. Evaluation of the performance of several gene prioritization methods.

Point-by-point response to comments:

Reviewer #1

R1.1 In general, the single cell ATACseq data of human heart is a potentially valuable resource. snRNA-seq of human heart tissue is readily available already. Previous published studies have provided candidate functional AF-associated SNPs, experimental validation of those SNPs, and candidate AF associated regulatory SNP target genes. The current study also provides such candidate SNPs and genes, but does not meaningfully assess the performance of their identification methods compared to these previous efforts, and does not provide experimental evidence that indeed functional SNPs and AF-associated genes were identified. Therefore, the selected SNPs and genes largely remain theoretical candidates.

We agree that it is important to compare our results with the published studies. We have now added additional comparisons with the most related studies. We summarize these comparisons here:

- Van Ouwerkerk et al, Identification of atrial fibrillation associated genes and functional non-coding variants, Nat. Comm., 2019. This study used a very loose criterion, of association $p < 10^{-6}$, to define candidate AF risk variants. The vast majority of these variants are probably non-causal variants. The authors then searched for genes in very large

regions (1.9Mb) near these variants. This gave a very large list of candidate genes. The authors then used various data, PC-HiC, eQTL, gene expression, to nominate ~300 putative risk genes. The study also overlapped putative GWAS variants (at a very loose cutoff of $p < 10^{-4}$) with putative regulatory elements in heart, obtaining 876 variants. As one can see here, the study relied on extremely loose cutoff (according to the GWAS standard) to define putative AF variants, lacking statistical rigor of fine-mapping. Without concrete knowledge of causal variants, we think the gene discovery results of the study would also be prone to false positives. Indeed, in our benchmark study, we found that the gene list from this study likely has a high false positive rate (see below).

- Van Ouwerkerk et al, Identification of Functional Variant Enhancers Associated with Atrial Fibrillation, *Circ. Res.*, 2020. This study examined 12 AF-associated loci, and used STARR-seq to identify variants with regulatory effects. Overlapping this list with GWAS data led to a list of 24 variants. With only 12 loci studied, this work, apparently, has a much narrower focus than ours. Similar to the previous work, the study also lacked rigorous attempts to identify causal variants. Indeed, we found that often the variants with allelic effects in STARR-seq have much lower association statistics, compared to our fine-mapped variants in the same loci. Statistically, these variants are unlikely to be causal variants of AF: most variants were not included in the “credible set” in fine-mapping, a commonly used way of narrowing down putative risk variants in GWAS loci.
- Hocker et al, Cardiac cell type-specific gene regulatory programs and disease risk association, *Science Adv.*, 2021. This study collected single-cell RNA-seq and ATAC-seq data in the human heart. It then used a simple fine-mapping procedure to nominate risk variants, and overlapped them with the open chromatin regions in cardiomyocytes (CMs). While this study is closely related to ours, its fine-mapping work did not take advantage of functional information. As a result, the study found few high confidence variants. Using a threshold of $PIP > 0.5$ (the same we used), the authors reported only 5 variants in open chromatin regions (Table S19 of the paper), compared to 42 from our work. The study also did not systematically nominate putative AF risk genes.

These comparisons have now been added to the manuscript. We have included Van Ouwerkerk 2019, in our gene discovery benchmark study. We have added a paragraph to compare with Van Ouwerkerk 2020, in the section, “Open chromatin regions in CMs are enriched with risk variants of heart diseases and inform statistical fine-mapping” - see the paragraph starting with “A recent study nominated putative causal variants in 12 AF-associated loci”. We revised our Discussion (second paragraph) to discuss the Hocker 2021 study.

About the comment that our study did not have “experimental evidence” of the functional SNPs, we’d like to point out that we did include results from a reporter assay. However, given the limited scope of our study in this regard, we were only able to test six high confidence variants. Among the 4 variants showing activities in a mouse cell line, 3 displayed allelic reporter activity (Fig. 4j).

While this is not a comprehensive validation, we believe this result provided some support to our conclusion that high PIP SNPs are likely functional variants.

Experimentally testing our predicted genes would be even harder. So we performed an *in silico* benchmark study. This type of *in silico* assessment has been commonly used in literature to assess gene discovery methods, e.g. Ndungu et al, AJHG, 2020 [PMID:31978332], and Nasser et al, Nature, 2021 [PMID: 33828297]. Please see the overview part of this letter for details.

R1.2 The correlation between cell type-specific OCR and expression (Extended Data Fig. 3d) is much higher than the correlation normally found. As accessibility is only one aspect of transcriptional activity, and quantitatively does not strongly correlate with expression level, the observed high correlation is remarkable. Please specify how the correlation was made, which parameters were used, whether quantitative data was used (level of expression, level of accessibility signal from ATACseq).

We apologize for omitting any description of this procedure and have added a detailed description to the methods section. The reviewer is right that accessibility of individual OCRs usually does not correlate highly with the transcript level of their presumed target gene. The high correlation in our analysis is due to the following reasons:

(1) The method we employ calculates a ‘gene score’, aggregating chromatin accessibility information within a window surrounding each gene (using the framework employed by ArchR, PMID:33633365). The score reflects the total effect of all open chromatin regions near a gene on its transcription. For example, the peaks near TSS would be weighted more in the calculation of the score. As such, the gene score metric has been shown to increase correlation between chromatin accessibility and transcript levels. It has also been used as the basis for integration of RNA and ATAC-seq data sets (Granja *et al.* "ArchR is a scalable software package for integrative single-cell chromatin accessibility analysis." *Nature Genetics*. 2021, Stuart *et al.* "Single-cell chromatin state analysis with Signac." *Nature Methods*, 2021).

(2) We calculated correlation scores in the following manner: First, we selected genes that were up-regulated in each cell type according to differential expression analysis of snRNA-seq data. Approximately 3000 genes were identified across all cell types. For each gene, the ATAC-seq gene scores and RNA-seq transcript counts, respectively, were aggregated across all cells in each cell-type cluster, followed by a log transformation. We then used these log-transformed pseudo-bulk gene scores and normalized expression levels to calculate Pearson correlation between gene scores and expression across cell types.

The resulting correlations are indeed high, providing strong support that both snRNA-seq and scATAC-seq clusters identified corresponding cell types.

R1.3.1 The fine mapping method of AF SNPs is not entirely clear to me. The authors focus on almost 900 highly AF associated SNPs. Previous work has indicated that also the large number of sub-threshold SNPs within associated variant regions and in LD with the highly significantly associated SNPs can be functional/affect regulatory element function (Wang et al, eLife 5, 2016). As these were not included in this analysis, a large number of causal SNPs are likely to remain undetected. Indeed, only 54 SNPs were marked to be probably functional in this study. Furthermore, the fine-mapped SNPs should be experimentally validated.

We acknowledge that in our submitted manuscript, we provided insufficient details of our fine-mapping analysis. We now clarify the issues raised by the reviewer. Our fine-mapping work followed the general strategy used in the literature. Specifically, fine-mapping analysis takes into account all variants in a region, regardless of their association p-values. We used large regions in our analysis, average 1-1.5Mb, that include all potential variants in LD. Fine mapping analysis then tries to identify one or a small number of causal variants that explain the entire association pattern in the region. It reports a posterior inclusion probability (PIP) for each variant in the region. Under some assumptions, PIP of a variant can be interpreted as the probability that the SNP is a causal variant of the trait (e.g. Schaid et al, “From genome-wide associations to candidate causal variants by statistical fine-mapping”, *Nature Reviews Genetics*, 2018).

This means that all variants in AF-associated loci (397,375 SNPs at $MAF > 0.05$) were included in our fine-mapping analysis, obviously including all sub-threshold variants. About the comment “only 54 SNPs were marked to be probably functional in this study”: in the updated results, we have 64 SNPs. Whether 54 or 64, we think this is actually a quite large number. Because of high LD in the human genome, in most fine-mapping studies, relatively few variants reach high confidence. To put our work in context, we cite some numbers from two published fine-mapping studies:

- (1) Mahajan et al, Fine-mapping type 2 diabetes loci to single-variant resolution using high-density imputation and islet-specific epigenome maps, *Nature Genetics*, 2018. With >800K samples of T2D patients and controls, the authors identified 51 variants at $PIP > 0.8$.
- (2) Ulirsch et al, Interrogation of human hematopoiesis at single-cell and single-variant resolution, *Nature Genetics*, 2019. The authors performed a fine-mapping analysis of GWAS of 16 blood cell traits in UK Biobank. The study identified 818 variants at $PIP > 0.8$ across all traits, or on average 51 per trait (Table S1 of the paper).

The numbers of high confidence variants from these studies are generally in line with our findings. Regarding the comment, “the fine-mapped SNPs should be experimentally validated”, we have validated a small number of high PIP variants (see response above to R1.1). Also, to put our work in context, systematic experimental testing of fine-mapped variants would be usually a focus of an

independent study. The two cited papers above, for example, performed computational fine-mapping and then mostly *in silico* analysis on the fine-mapped variants.

We have added additional text in our manuscript to provide more background of statistical fine-mapping (the first section of Results, “Overview of the experimental and computational approach”). We also revised the relevant section in Methods (titled “Fine-mapping causal variants in AF-associated loci”), with the details of general fine-mapping analysis, and the specific approach we used to incorporate functional information to improve fine-mapping.

R1.3.2 Also, the results should be compared with those of previously published efforts to identify causal AF associated SNPs such as van Ouwkerk et al., Circ Res 2020 using STARR-seq of a number of AF associated loci.

We agree with the reviewer that it is important to compare our results with previously published efforts. See our response to R1.1 above for this comparison.

R1.4 The fine mapping of SNPs seems to rely on the OCRs identified in the scATACseq data from the clustered ventricular cardiomyocytes. Something very similar has been done already in van Ouwkerk, Nature Communications 2019. In this study, ATACseq data from purified human atrial cardiomyocytes was used to identify AF associated SNPs (including the sub-threshold ones) that co-localize with atrial cardiomyocyte-specific OCRs and with human cardiac enhancers. The authors should compare their results with the mapping in this study, provide arguments how their approach is complementary, how it performs compared to this and other studies mapping AF SNPs, and provide experimental evidence that their fine mapping indeed enriches for functional SNPs.

We agree this is a very related study. Please see our response above to R1.1. We think the approach used in the study, overlapping GWAS variants at a loose cutoff with some functional annotation (enhancer data), has significant limitations compared with the more rigorous fine-mapping approach. We listed a few here: (1) This analysis may nominate a large number of variants, most of which are unlikely causal variants of the phenotype. The referred study, nominated 876 variants (Fig. 6e and Supplementary Data 5), on average ~8 variants per locus. (2) This approach ignores important quantitative information when comparing nominated variants. For example, a variant with $p = 10^{-6}$, and a variant with $p = 10^{-10}$ in GWAS, both residing in enhancers, would obviously carry very different strength of evidence in terms of being causal variants. More generally, it is unclear how one would compare, for example, a variant with strong association, e.g. $p = 10^{-10}$, but not in a regulatory region, vs. a variant with weaker association, e.g. $p = 10^{-6}$, but inside a regulatory element.

It is for these reasons that fine-mapping has been increasingly used as a more rigorous way of risk variant discovery in GWAS. In addition to the two studies cited above (response to R1.3.1), we provide a few more studies as examples here:

- Weissbrod et al, Functionally informed fine-mapping and polygenic localization of complex trait heritability, Nature Genetics, 2020
- Fachal et al, Fine-mapping of 150 breast cancer risk regions identifies 191 likely target genes, Nature Genetics, 2020
- Ulirsch et al, Interrogation of human hematopoiesis at single-cell and single-variant resolution, Nature Genetic, 2019
- Schwartzenuber et al, Genome-wide meta-analysis, fine-mapping and integrative prioritization implicate new Alzheimer's disease risk genes, Nature Genetics, 2021
- Zhang et al, Allele-specific open chromatin in human iPSC neurons elucidates functional disease variants, Science, 2020 (from our own work)

R1.5 It was not clear to me which SNPs were used to identify the AF risk genes. Did the authors use the small set (of 54) fine-mapped SNPs? If so, this analysis could be biased by the stringent but un-validated selection method, leading to under-reporting of candidate genes. Moreover, while the authors compared their results with those of Nielsen et al and van Ouwerkerk et al., they do not draw any conclusions from this comparison, and do not provide insight into the relative performance of the three methods to identify AF-associated variant target genes. Again, experimental validation of their candidates is lacking.

To clarify, our fine-mapping uses all variants in AF associated loci, as discussed above (response to R1.2). After fine-mapping, our gene discovery procedure, Mapgen, uses all variants that have some plausibility of being causal variants ($PIP > 10^{-5}$). Mapgen then links putative target genes of each variant, and then aggregate information of all the variants to come up with scores for all genes. We have revised our text in the section “A novel computational procedure utilizes fine-mapping results to identify AF risk genes” (the first paragraph), as well as Fig. 5a, to better describe the intuition of the method.

We agree with the reviewer that the true causal variants may not be the top GWAS variants, or even have sub-threshold associations. *One of the key advantages of Mapgen is that it uses information of all variants, including those with weaker or sub-threshold associations.* As an example, we show the result of one locus below. Fine-mapping at this locus suggests several putative causal variants, but without high confidence on each variant (max. $PIP = 0.215$). Because all these variants are either close to GJA5, or looped to the promoter of GJA5, Mapgen nominated GJA5 as the potential risk gene with high confidence (gene $PIP = 0.802$). GJA5 is a well-known AF risk gene (Gollob MH, *et al.*, Somatic mutations in the connexin 40 gene (GJA5) in atrial

fibrillation, *N Engl J Med*, 2006). This example and another one (NKX2-5 locus) are now added in the manuscript (Fig. 5d, Extended Data Fig. S12).

Figure 2. Fine-mapping and Mapgen analysis of a locus containing GJA5. The top two tracks represent the $-\log_{10}$ p-value of SNPs from AF GWAS (with color representing LD with the lead SNP) and their PIPs from SNP-level fine-mapping. Middle tracks represent cell-type aggregated ATAC-seq signals (CM: red, endothelial: green; fibroblast: purple), followed by heart H3K27ac and fetal DHS peak calls. The last track shows interactions identified from promoter-capture HiC data in iPSC-derived CM. Vertical bars show the locations of the four fine-mapped SNPs supporting GJA5 as the risk gene in this locus. The red links in the PC-HiC track show interactions linked to these four fine-mapped SNPs

As we have mentioned in our response to R1.1 above, we have provided validation of Mapgen, in comparison with van Ouwkerk et al, *Nat. Comm*, 2019.

R1.6 the section on eQTLs (lane 333 and further) was difficult to follow for me. Please consider clearer description and/or a figure explaining the topic using an example.

We, too, realized that the eQTL section in the submitted manuscript was difficult to follow. We have largely revised the beginning part of this section, explaining the motivation and rationale of

our analysis. Briefly, we switched our focus to understanding why most of the AF variants we discovered showed no evidence of acting as heart eQTLs. Given that eQTLs are widely used to interpret GWAS findings, we think this is an important problem and of general interest to the audience.

Our main hypothesis tested is that bulk eQTLs are best at detecting regulatory variants with broad effects across multiple cell types, and will miss a large number of variants with effects limited to certain cell types. Our key findings are:

- We inspected the eQTL sharing pattern from multiple tissues in GTEx. The majority of heart eQTLs are highly shared across a diverse set of tissues, with <10% eQTLs found in 5 or fewer tissues (including heart). Given that CMs are not widely shared across different tissues, this suggests that most heart eQTLs are unlikely to be CM-specific.
- Among heart eQTLs, we found that the ones within CM-specific OCRs are depleted, compared to eQTLs in OCRs shared across multiple cell types.
- We used simulations to explore how the power of detecting cell-type specific eQTLs depends on the proportion of that cell type in the tissue. We found that even when the cell type is 20-30% of the tissue, as is the case for CMs, the power drops dramatically.

Together we think these results shed light on the limitation of bulk eQTLs in detecting cell-type specific regulatory variants, and in interpreting GWAS findings.

Reviewer #2 (Remarks to the Author):

The manuscript by Selewa, et al describes the use of single-cell chromatin accessibility data in cardiac tissue to improve on the fine-mapping of putative casual variants and their target genes at GWAS loci for atrial fibrillation (AF). They also investigate the challenges of detecting of eQTL in bulk tissue studies. The primary contributions are the novel adaption of Bayesian fine mapping to include a prior based on information from the single-cell chromatin data, and the novel method to determine target genes for fine-mapped variants. Though there is a focus on AF, it isn't as clear whether specific proposed casual variants and/or target genes represent a significant contribution to understanding this disease. Validation experiments do confirm the effects of proposed genetic variants on regulatory activity. The results presented provide evidence that these proposed new methods can help in better determining how genetic variation may be contributing to phenotypes through altered gene regulation.

My primary concern and difficulty in evaluating the results presented are that while these new methods demonstrate characteristics desired in improved fine mapping (i.e. confidence measures indicating a greater degree of accuracy, fewer variants in credible sets) and in target gene prediction (distributed evidence, evidence of function in disease-related processes), there is no real

empirical evidence showing that the predictions are accurate or at least more accurate than other methods. Understandably, performing functional experiments on predicted variants/genes to more definitively show mechanism are difficult, but either arguments for why these methods are more accurate than something else need to be made clearer, or experiments (could be analytical) need to be designed to provide more evidence. Some more specific comments follow, not in order of importance.

We thank the reviewer for these comments and agree that a more careful assessment of our procedure is important. About the advantage of our method for variant discovery, we have provided more background of fine-mapping analysis in the method overview section (the first section of Results). We copied it here:

“Fine-mapping is a technique that aims to identify one or few causal variants that explain all the associations in a locus. It avoids the use of arbitrary LD cutoffs in selecting candidate variants, and is able to quantify the uncertainty of each nominated variant. Recent fine-mapping techniques are also able to incorporate functional information of variants, such as regulatory activities in trait-related cell types. Because of these benefits, fine-mapping techniques have been successfully applied to many common traits such as Type 2 Diabetes, Schizophrenia and autoimmune disorders.”

Given the limited space, it is impossible to discuss all the conceptual advantages of fine-mapping vs. heuristic ways of choosing putative variants, e.g. those based on LD with the lead variants. We listed here just a few papers that have used statistical fine-mapping to study genetics of complex traits (all cited in the manuscript):

- Ulirsch et al, Interrogation of human hematopoiesis at single-cell and single-variant resolution, *Nature Genetic*, 2019
- Fachal et al, Fine-mapping of 150 breast cancer risk regions identifies 191 likely target genes, *Nature Genetics*, 2020
- Weissbrod et al, Functionally informed fine-mapping and polygenic localization of complex trait heritability, *Nature Genetics*, 2020
- Schwartzenuber et al, Genome-wide meta-analysis, fine-mapping and integrative prioritization implicate new Alzheimer’s disease risk genes, *Nature Genetics*, 2021
- Zhang et al, Allele-specific open chromatin in human iPSC neurons elucidates functional disease variants, *Science*, 2020 (from our own work)

We also revised the relevant section in Methods (tilted “Fine-mapping causal variants in AF-associated loci”). We have added the details of general fine-mapping analysis, and the specific approach we used to incorporate functional information to improve fine-mapping.

To demonstrate more clearly the benefits of our gene discovery method, Mapgen, we have revised the explanation of this method in the method overview section (the first section in Results), and the first paragraph of the section “A novel computational procedure utilizes fine-mapping results to identify AF risk genes.” More importantly, we have benchmarked the performance of the Mapgen showing that it performs better than other methods (see the beginning section of this letter).

R2.1 There are at least two other single-cell chromatin studies in heart tissue. One is the recently published human atlas (<https://pubmed.ncbi.nlm.nih.gov/34774128/>) and one more focused on cardiac cells (<https://pubmed.ncbi.nlm.nih.gov/33990324/>) that also investigated AF. How do the cell type clusters in the current study compare to those? Do you find essentially the same clusters, or are there any discrepancies? Is there significant agreement in which regions are considered accessible in like cell groups? These are focused on establishing how consistent single-cell ATAC is across multiple studies, and thus generalization of results presented here, as well as supporting the accuracy of the cell clusters.

We thank the reviewer for these comments and now include an in-depth comparison with the data from Hocker et al.. We chose this dataset since it had significantly higher coverage of cell types from the heart and the other study mentioned by the reviewer used a subset of the samples from this study for their heart sample.

We found essentially the same cell types in our snRNA-seq data and scATAC-seq data as reported in Hocker et al. (Figure 3 below).

[redacted]

Figure 3: Comparison of cell types from scATAC-seq analysis in our study and Hocker *et al.* UMAP on the left taken from Hocker et al., UMAP on the right from our manuscript.

One difference is that the Hocker et al study profiled both atrial and ventricular samples, while our study profiled only samples of ventricles (as well as samples from the interventricular septum and apex). As a result we did not detect atrial cardiomyocytes. There is another small difference between the two studies. While we identified Pericytes and smooth muscle cells as two separate but similar clusters in both our snRNA-seq and scATAC-seq data, Hocker et al did not label any cells as Pericytes in their scATAC-seq and instead assign all cells within the broader area to smooth muscles. However, they did detect pericytes in their snRNA-seq data.

We are confident in our scATAC-seq annotation because the computational assignment of cluster labels using our snRNA-seq data as reference was clear (Extended Data Figure 3b). Additionally the cells within clusters designated as smooth muscle and pericytes show higher locus-specific accessibility for smooth muscle markers (e.g. MYH11) and pericyte markers (e.g., KCNJB8), respectively (Extended Data Figure 4). Given that Pericytes and smooth muscles are closely related, the ambiguity is less surprising. Regardless of annotation, we found no association of Pericytes and smooth muscles with AF variants.

We describe this comparison in the manuscript, see the first paragraph of the section “Single-cell transcriptome and chromatin accessibility profiling reveals multiple cell types in the human heart”:

“We also found good agreement between cell types identified in our scATAC-seq data and a recent study (Hocker et al, "Cardiac cell type-specific gene regulatory programs and disease risk association.", Sci Adv., 2021), the only difference in annotation between these two studies was that we detected separate pericytes and smooth muscle clusters (Fig. 2a), whereas Hocker et al. annotated a single large cluster as ‘smooth muscle’. These results supported our cell-type assignments in both modalities.”

About the comparison of specific regions accessible between two studies, see our response to R2.2 below.

R2.2 Similar to the above, OCRs are compared with bulk tissue DHS, but why not also compare with the existing single-cell studies?

We have now added a comparison of our OCRs with peaks from Hocker et al. and included it as Extended Data Fig. 7 (shown as Figure 4 below). Specifically, we compared the overlap between peak calls from each of their cell type clusters with peaks (OCRs) detected in matched cell types from our data. We find good agreement between peak calls in these two studies reaching around 70% peaks overlapping for any given dataset (with exceptions associated with smaller clusters).

We consider this overlap fairly high given that data were generated from different individual samples, in different laboratories using different pipelines and experimental setups.

Figure 4. Comparison of peaks (OCRs) identified by Hocker *et al.* and this study. The numbers of peaks in the overlapped areas of the Venn Diagrams are from our study. The numbers of overlapped peaks from Hocker *et al.* may differ slightly because of the fact that a peak in one dataset may overlap two in the other. The numbers, however, are generally similar.

R2.3 The previous study on cardiac cells (<https://pubmed.ncbi.nlm.nih.gov/33990324/>) also reported the enrichment of AF risk variants in cardiomyocytes. This should at least be acknowledged.

We thank the reviewer for pointing this out. We have added a new paragraph in the Introduction, reviewing several papers in literature, including the one mentioned (Hocker *et al.*), that used functional/epigenomic data to help find risk variants of AF. We also cited the results of Hocker *et al.* when we presented our finding that AF risk was enriched in cardiomyocytes (first paragraph in the section “Open chromatin regions in CMs are enriched with risk variants of heart diseases and inform statistical fine-mapping”). We also added a comparison of the fine-mapping results of Hocker *et al.* vs. ours in the Discussion. We copied the relevant text here:

"Compared with several recent studies that aimed to identify risk variants and genes in AF (van Ouwerkerk *et al.* and Hocker *et al.*), our study has a few key advantages. Hocker *et al.* intersected fine-mapped variants with cell-type-resolved OCRs to nominate putative regulatory variants. Their work and related studies (Chiou *et al.*, Single-cell chromatin accessibility identifies pancreatic islet cell type- and state-specific regulatory programs of diabetes risk, *Nature Genetics*, 2021) demonstrated the utility of single cell ATAC-seq data for interpretation of non-coding variants from GWAS. Our work extends these studies by using a computational procedure that leverages the strong enrichment of genetic signals in CM-specific OCRs to fine-map causal variants, identifying 68 high confidence SNPs at PIP > 0.5, including 42 in CM OCRs, compared to five nominated by Hocker *et al.* (Fig. 4b)."

R2.4 PR interval first mentioned on line 164, but is not defined and it's not clear why this phenotype is even mentioned since it is not really discussed in any detail.

We included the PR interval because the trait is related to cardiac rhythm. We added a sentence in the relevant part of the text (first paragraph in the section “Open chromatin regions in CMs are enriched with risk variants of heart diseases and inform statistical fine-mapping”).

“Interestingly, the variants associated with the PR interval showed a similar enrichment pattern, suggesting a genomic link between PR interval and AF risk for future investigation (Fig. 4a).”

R2.5 Using a prior to favor variants in certain regions is shown to increase the number of high confident variants ($PIP \geq 0.5$). But is this simply a result of having a prior and not necessarily the prior providing useful additional information? What if you randomly assigned priors? Or assigned them based on distance to the reported lead SNP? It would be good to better show that your specific priors are contributing in a unique way, especially due OCR information.

We first point out that using prior information to improve fine-mapping is a strategy used in previous studies. This strategy is based on “Empirical Bayes”, a statistical framework to incorporate prior information in statistical inference. Briefly, the prior information is not “subjective” input from the data analysts, instead it is estimated from analysis of real data. For example, if using genome wide data, we found that putative risk variants are 10 times enriched in OCRs, then in fine-mapping any specific locus, the prior probability of a variant in an OCR would be set 10 times higher than the background. We cited a few papers here that used the Empirical Bayes strategy to incorporate functional information in fine-mapping:

- Pickrell, Joint analysis of functional genomic data and genome-wide association studies of 18 human traits, *Am. J. Hum. Genet.*, 2014
- Weissbrod et al, Functionally informed fine-mapping and polygenic localization of complex trait heritability, *Nature Genetics*, 2020
- Zhang et al, Allele-specific open chromatin in human iPSC neurons elucidates functional disease variants, *Science*, 2020 (from our own work)

To show that functional information (OCRs in our case) actually improved fine-mapping, we had included a comparison of the fine-mapping results with functional prior vs. those with uniform prior (i.e. treating all variants equally), see Fig. 4b. We found 68 candidate variants at $PIP > 0.5$ using the functional prior vs. 44 variants under the uniform prior. Upon the suggestion of the reviewer, we have performed a few additional analyses using different priors. In the first, we used a “shuffled” prior in fine-mapping, where the functional priors learned from data are randomly shuffled among the SNPs. In the second, the “gene body prior”, the variants within the gene bodies have prior probabilities 10 times higher than the background. In the third, “TSS prior”, the variants within 20 Kb of TSSs of genes have prior probabilities 10 times higher. In all cases, we found that using the functional priors led to more high PIP variants (Figure 5).

Given that fine-mapping with functional prior has been used quite often in the literature, we feel that it’s sufficient to show the comparison with the uniform prior in the text. So we did not include in the text comparison with other types of priors we investigated here.

Figure 5. Distribution of PIPs of variants using different priors. (a) Histogram of variants at different bins of PIPs. Each panel shows a comparison of the results from functional prior vs. another prior. Given that we are usually interested in only high PIP variants, only results under $PIP > 0.5$ were shown here. (b) The empirical cumulative distributions of PIPs in the range of $[0.1, 1]$ for all types of priors.

R2.6 You state that your fine-mapping method can help determine the cell types a variant may be most relevant in. Can you use your single-cell RNA-seq data to further support the mappings of variants to cell types? You give a couple of examples (IKZF3, TNFSF13) that based on their known functions agree with the cell type mapping, but can this be done in a more systematic way? Are the relevant OCR annotations for these variants enriched, say, around genes that are also cell-specific or differentially expressed in that cell type?

We thank the reviewer for this suggestion. We used differential expression data to assess the cell type partitioning result. We focused on cardiomyocytes here, as the majority of the loci have cell type specific signals in cardiomyocytes. Using the criterion PIP proportion $\geq 50\%$ in cardiomyocyte OCRs, we selected 46 loci where the causal variants are likely acting through CM. Among these loci, we were able to assign likely target genes in 27 loci, using the criteria of gene PIP ≥ 0.5 . Nine out of 27 genes (33.3%) are cardiomyocyte differentially expressed genes (DEGs). In contrast, only 29 out of 842 (3.4%) low PIP genes (gene PIP < 0.1) in these 46 loci are cardiomyocyte DEGs (Figure 6 below). We believe the nearly 10 fold enrichment of DEGs is a strong support of our approach to identifying the relevant cell types of individual loci. These results were now included as Extended Data Figure 13a.

A related analysis reported in the original manuscript also supported cell type specific expression of our candidate genes. We showed that high PIP genes tend to have higher expression in CMs than low PIP genes (Fig. 6a in the revised manuscript). We did not discuss two genes from the original submission, IKZF3 and TNFSF13, in the revised text. With the updated fine-mapping results, these two genes no longer reach PIP > 0.5 the threshold we used for inclusion.

Figure 6. Enrichment of CM DEGs in high PIP genes vs. low PIP genes, in loci where the proportions of PIPs in CM OCRs are $\geq 50\%$.

R2.7 HL-1 cells should be defined as a cardiac muscle cell line to make the relevance obvious. Figures for reporter assays should state how many replicates were used in creating the plots/error bars.

We have added the suggestion to the text and changed the figure legend to include the number of replicates.

“Four out of six variant-containing OCRs induced reporter gene expression in mouse cardiac cells (HL-1 cell line) (Extended Data Fig. 6a, Methods), but not in a fibroblast line (3T3), suggesting cell-type-specific activity of the four OCRs (Extended Data Fig. 6b).”

Figure legend: “Fig. 4j. Reporter activities in cardiac cells (HL-1) cells of regions containing selected SNPs, with both reference and alternative alleles. Data are from 5 replicates for each construct. p-values were calculated using a paired two-sided t-test.”

R2.8 You state that the lack of overlap between your fine-mapped variants and heart eQTL in bulk tissue suggests a limitation in of the eQTL – but why is this necessarily true and not that your fine mapping is wrong?

We think this low overlap is unlikely due to problems with fine-mapping. Firstly, we checked if the top GWAS variants in these loci are heart eQTLs. We limit to the loci with high PIP variants ($PIP > 0.5$). Only 19 out of 68 (28%) top GWAS signals are eQTLs. So the generally low overlap of eQTL and GWAS does not really depend on fine-mapping. Secondly, a GWAS variant may be associated with gene expression, but does not have a causal effect on expression. So we checked if our fine-mapped GWAS variants “colocalize” with eQTLs, which would suggest that the same causal variants affect both the phenotype and expression. We ran *coloc*, a widely used program for this purpose. The results show that, among 19 GWAS top SNPs that are also heart eQTLs, only four have evidence of colocalization ($PP4$ in *coloc* > 0.5). Again, the result here does not depend on fine-mapping.

Another notable fact about fine-mapping is: the biggest problem for fine-mapping is usually its low resolution, i.e. it cannot distinguish variants in high LD. In those cases, each SNP in LD would receive low PIP. But if fine-mapping does identify SNPs with high PIP, these results are generally credible. With some mild assumptions, most importantly that all causal variants are included in analysis, the PIPs would reflect the probabilities that the variants are causal.

Our results of low sharing among GWAS and eQTLs also do not come as a surprise. They add to the growing evidence that GWAS variants are often not colocalized with eQTLs, and that eQTLs mediate a relatively small fraction of the heritability of GWAS traits. We cite a few papers here:

- Chun et al, Limited statistical evidence for shared genetic effects of eQTLs and autoimmune-disease-associated loci in three major immune-cell types, Nature Genetics, 2017. This paper showed that in only ~25% loci of autoimmune diseases, the GWAS signals and the eQTLs from immune cells were driven by the same causal variants.
- Yao et al, Quantifying genetic effects on disease mediated by assayed gene expression levels, Nature Genetics, 2020. This paper developed a method to quantify how much disease heritability was attributed to eQTLs. Using GTEx eQTL data and a diverse range of traits, it estimated that, averaging across traits, only $11 \pm 2\%$ of heritability was mediated by assayed gene expression levels.

R2.9 You present your Mapgen method that you propose is more accurate at assigning target genes. But, you don't really present much evidence that this is more accurate than the simple method of assigning the closest gene. In fact, 39 of 45 of your predictions were the closest gene. Your main evidence for the other 6 is that 5 have annotations that show their involvement in the heart. But for these 6, are the closest genes also involved in heart function? You show one has PC-HiC evidence, which is really the only one that shows good evidence of accuracy. For rs1152591, why is uncertainty better? This is only true if all four potential genes are involved. The arguments for not using chromatin looping or eQTL are rather weak. While these may not be comprehensive, it's not clear how detrimental it would be to try and incorporate that information.

We think this is a fair criticism and we have added additional analyses in the revised manuscript. But firstly, we'd like to clarify the difference between Mapgen and the nearest gene method. The fact that 39/45 of our high PIP genes are nearest may give the impression that the two methods behave similarly. However, this is misleading. The comparison here only looks at the high PIP genes found by Mapgen. If we look at the entire result across all loci, as shown below, the results differ greatly between the two methods. With regard to why high PIP genes are often the nearest genes: this is due to the fact that the power of finding high PIP genes is higher in nearest genes. In calculating gene PIPs, Mapgen gives more weights to variants close to genes than distal variants. For Mapgen to give high PIPs to a distant gene, there must be some evidence, such as PC-HiC or ABC, that support the distal gene as a target of the causal variants. But as we discussed in the paper, often these datasets either do not link variants to any genes; or they link variants to multiple genes, in which case, none of the genes would receive high PIPs.

We performed an *in silico* benchmark study to compare Mapgen with other methods, including the nearest gene method. Please see the beginning of this letter where we summarized this new analysis.

We also updated our results in the submitted manuscript where we compared the high PIP genes from Mapgen vs. nearest. The results were summarized in a supplementary table. We copied the text below:

“Among the 46 genes at $PIP \geq 0.8$, eight (17%) were not the nearest genes, by distance to TSS, to the top GWAS SNPs. Some of these genes have been implicated in AF and related phenotypes, including KCNN3, TTN and HCN4. Most of the other genes have plausible functions such as CALU, SSPN, and PKP2 (Supplementary Table 11). Most of the nearest genes in these loci, in contrast, showed limited or no functional relevance (Supplementary Table 11). As an example, in the locus containing CALU, the nearest gene of the top SNP, rs55985730 (PIP 0.91) is OPN1SW, an opsin gene with function in color vision, but no clear relevance to AF. This SNP is looped to a distal gene CALU in PC-HiC data (Fig. 5f), allowing Mapgen to identify CALU as the likely risk gene. CALU is a calcium-binding protein and involved in alleviation of endoplasmic reticulum (ER) stress in cardiomyocytes [PMID:24012670]. ER stress has a critical role in the pathophysiology of AF [PMID:35058801]. These results suggest that by using chromatin loop information, Mapgen is able to identify distal risk genes.”

About the comment regarding rs1152591: in the revised text, we have removed this example.

About the comment that, “The arguments for not using chromatin looping or eQTL are rather weak”. This may be due to a misunderstanding of the text. Mapgen does use chromatin loop information in linking variants to genes. See the CALU example discussed above. What we said is that using chromatin loop information alone, would lead to low power of gene discovery, as we show in the Figure 1 at the beginning of the letter (ABC-max method). Regarding eQTL information: we have not used eQTLs in the current analysis for two reasons. First, we found limited overlap of fine-mapped variants with the heart eQTL data as discussed above (response to R2.8). Second, even if a variant, say G, is an eQTL of some gene X, it does not necessarily mean that X is the target of the variant G. It is possible that G is LD with the true causal variant of X expression, and G itself has no effect on X. Nevertheless, it is possible to include eQTL information in some way in Mapgen, and that is something that our research group plans to work on.

R2.10 There is a repeated claim that sharing of eQTLs across tissues is “surprising” or “puzzling”, which I’m not sure reflects a general consensus in the community. Admittedly, the current studies in tissues do find a certain amount of overlap across tissues, but is the amount really surprising? What was the prior expectation? The issue of tissue heterogeneity is well-known, and in the latest GTEx analysis, they specifically attempted to investigate the influence of this. In their study, they found “the pairwise relatedness of GTEx tissues derived from their cell type composition is highly correlated with tissue sharing of regulatory variants (cis-eQTL versus cell type composition Rand index = 0.92)”. This suggests that common eQTL in tissues are due to common eQTL in underlying cell types, thus this idea that there is a lack of awareness that performing eQTL on tissues misses underlying eQTL in specific cell population seems wrong. The analysis does nicely

present some observations about the effect of performing tissue eQTL, but I think the significance of these observations is a bit over-stated.

We thank the reviewer for pointing out this very relevant observation from GTEx analysis. We also agree that because of shared cell types among tissues, it is not surprising that eQTLs of these common cell types would be shared across tissues. We have removed our earlier claim that sharing of eQTLs across tissues is “puzzling”. We also feel that overall, this section was somewhat disconnected from the rest of the paper. So we largely rewrote the introductory part of this section, with a focus on a somewhat different question, why most of our fine-mapped variants of AF are not heart eQTLs. Our hypothesis is that the bulk heart eQTL study missed a large fraction of regulatory variants specific to CMs, thus were depleted of AF risk variants.

We think it is interesting to test this hypothesis. This is not a simple corollary of the observation that shared cell types may drive shared eQTL across tissues. In fact, a significant fraction of cells (20-30%) in the heart are cardiomyocytes (CMs), which are not shared widely across tissues. So we would expect that a substantial fraction of heart eQTLs would be limited to a small number of tissues, including the heart. This is not what we observed. Only 10% of heart eQTLs are limited to <5 tissues. So we performed quite a few analyses in this section to help answer this question.

Our key finding is about the power difference of detecting cell-type-specific eQTLs, vs. in detecting eQTLs with shared effects across cell types. It might sound intuitive or even trivial, that it would be easier to detect eQTLs in shared cell types. But the reality is probably more complicated. Imagine we have an eQTL of a gene acting only in one cell type, and the expression of that gene is largely limited to that cell type, then the power of detecting this eQTL would be similar, regardless of the proportion of that cell type in the tissue (whether it's 10% or 90%, say). The intuition is that, the expression of the gene in bulk RNA-seq, would mostly come from just one cell type, so even if the data is from bulk tissues, we can think of it as from purified cell type. This simple hypothetical example suggests that there is actually a complicated relationship between eQTL detection power and a number of factors, including the cell type proportion, gene expression level, how gene expressions are correlated across cell types in the bulk tissue, and so on. We touched upon this issue in our simulation and Supplementary Notes.

R2.11 In Figures where abbreviations for cell-types are used as labels, these should be defined in the figure legends.

This has been changed.

Reviewer #3 (Remarks to the Author):

With the increasing availability of functional genomic data at single-cell resolution, integrative analyses of single-cell genomic data, particularly epigenomic data, with GWAS is important for interpreting the function of genetic variants and defining the etiology of disease. This study profiles both single-cell transcriptomic and epigenomic data in the adult human heart, which can be a useful resource. The new computational procedure named Mapgen has the potential to be scalable to other studies. Overall this is a valuable paper that should be published, but a few concerns need to be addressed before publication, as I note below.

We thank the reviewer for the supportive comments and address the raised issues below.

Major Comments:

Data quality and accessibility:

R3.1 The cell type specific marker genes from the snRNA-seq data appear lowly expressed as shown in Fig 2c. Can more validation of the cell type specific clusters be performed to bolster this further?

We have added additional analysis that supported our previous cell type annotations. We used previously published Heart Cell Atlas (HCA) data from Litvinukova et al. (Litviňuková, M., et al., Cells of the adult human heart, *Nature*, 2020) as reference and transferred their labels onto our snRNA-seq data, using Azimuth (Hao, Y et al., Integrated analysis of multimodal single-cell data. Cell 184, 3573-3587.e29). This step effectively matches the cells in our data with their closest neighbors in the HCA data, and then uses the provided cell type labels of the HCA cells to label the cells in our data. We found that the cell type annotations from this label-transfer analysis are highly consistent with our own cell type annotations (Figure 7a below).

While the marker genes we previously showed are in some cases lowly expressed, they are nevertheless very specific for the clusters. We have updated our figure of marker gene expression (updated Fig. 2c, and Figure 7b below). To better visualize, we also showed the percent of nuclei expressing a certain marker gene in Figure 7c below.

Figure 7. (a) Comparison of cell-type annotation in our study to labels transferred from Litvinukova *et al.*. Rows are cluster labels in our study with the number of cells indicated for each cell type in brackets. Columns are labels from Litvinukova. Shown are the proportions of cells per cluster (row) that are annotated with a given label in Litvinukova *et al.* (b) Stacked violin plots of marker gene expression (log-normalized expression values) in each cell type. (c) Proportion of nuclei expressing a marker gene across all clusters.

R3.2. It is surprising that most open chromatin regions (OCRs) with cell-type-specific accessibility are enriched in intronic regions (Fig. 3b), although a high enrichment in the non-coding regions makes sense. Is this observation consistent with prior studies? If not, could the author discuss why this may be the case.

There is indeed a modest enrichment of OCRs in introns. We checked another published single-cell ATAC-seq dataset in heart (Hocker *et al.*, PMID:33990324), and found similar patterns (Figure 8 below). In both studies, this enrichment is also more pronounced in CMs compared to other cell types: 47.76% of CM-OCRs are located in introns vs. around 40% of OCRs in other cell types are found in introns. While it is unclear what exactly contributes to intronic enrichment of OCRs or

its functional relevance, we observed a large number of peaks and generally accessible chromatin in the gene bodies of highly transcribed genes (e.g. Fig. 2b and Extended Data Figure 4).

Figure 8: Genomic distributions of the OCRs in this study (blue), compared with those from Hocker et al. (red). The distribution was computed by *calcPartitionsRef* function in GenomicDistributions R package (Kupkova et al. BMC Genomics. 2022). Only OCRs in Cardiomyocyte, Endothelial and Fibroblast clusters are shown here.

R3.3. Given that the numbers of cell type-specific peaks across different cell types are not the same, it does not appear to be valid to just show the proportions of overlap with bulk level data (Fig. 3d). The odd ratios are useful to correct differences in peak numbers. However, it would be helpful to show how many informative peaks are exclusively identified by single-cell data.

We agree with the reviewer that just showing the proportions of overlap does not give the full picture. We have performed the analysis as suggested, comparing our peaks vs. ENCODE DHS peaks and H3K27ac peaks in the heart. For each comparison, we assessed the enrichment of peaks from one dataset in the other dataset, measured as odds ratios (ORs). This analysis shows a very strong enrichment of peaks in relevant cell types (Figure 9 below, included in the manuscript as Extended data figure 6).

Figure 9. Comparison of open chromatin regions between the current study and ENCODE DNase hypersensitivity sites (DHS) from the indicated tissues (a) and ENCODE H3K27ac peaks in the heart (b).

We have now updated the relevant part. We decided to keep the original heat-map in the submitted manuscript, given that the proportions of overlap are easier to interpret, while ORs are somewhat less intuitive. We have included the OR-heatmap in the supplement (Extended Data Fig. 6).

R3.4. The scRNA-seq and scATAC-seq generated by this study is a valuable resource. The processed data should be made publicly available and ideally a reviewer code would have been provided, as currently it is only listed as something that “will be deposited to the Gene Expression Omnibus.”

We agree with the reviewer and apologize for the delay in depositing the data. The data have been generated as part of the HCA and a prime objective of this effort is sharing of the data. The data has been deposited in GEO repository: accession GSE224997.

Functional fine mapping and interpretation:

R3.4.1. Functionally informed fine-mapping is performed by using an existing method called SuSiE. The use of this method should be discussed explicitly in the results section rather than only mentioning this in the methods and methodologic credit should be provided.

We completely agree. We have now explicitly mentioned the name of the methods we used for fine-mapping analysis, including TORUS and SuSiE, with the relevant citations. We copied the text below in the section: “Open chromatin regions in CMs are enriched with risk variants of heart diseases and inform statistical fine-mapping” (the second paragraph)

“This observation motivated us to statistically fine-map causal variants in 122 approximately independent AF-associated loci. We first used TORUS (Wen et al., AJHG, 2016) to estimate how putative risk variants are enriched in multiple functional annotations, including protein-coding regions, conserved sequences, and OCRs in CMs (Extended Data Fig. 9a, Methods). This information was used to set prior probabilities of variants being causal. We then performed fine-mapping analysis of all AF-associated loci with SuSiE (Wang et al., The Journal of the Royal Statistical Society, Series B (Statistical Methodology), 2020).”

R3.4.2. The author use fine mapping by integrating OCRs. Assigning priors to variants according to OCRs for fine mapping can increase the interpretability (Fig. 4b). It is important to show how many loci will be missed by this approach compared to traditional fine mapping? The author can illustrate chromatin accessibility and additional functional genomic annotations that are similar to Fig. 4h but for SNPs called from traditional genetic fine mapping?

In our manuscript, we showed a scatter plot (Fig. 4b) comparing the PIPs of all variants from our functionally informed fine-mapping vs. those from traditional fine-mapping (uniform prior). While a few variants from uniform prior were lost in our results, we gained many more sites. We have summarized the comparison of the two sets of results below (Figure 10a). We have now added this figure into Extended Data Fig. 9.

As suggested, we annotated the SNPs at $PIP > 0.5$ from the traditional fine-mapping using the genomic information. These results are shown below (Figure 10b). In general, we found that a lower fraction of SNPs here were associated with regulatory annotations.

Figure 10. Comparison of fine-mapping results from functionally informed fine-mapping vs. those from traditional fine-mapping with uniform prior. (a) The number of SNPs with PIP ≥ 0.5 identified two types of priors. (b) Chromatin accessibility and additional functional genomic annotations of 44 SNPs (PIP $\geq 50\%$) identified by traditional fine-mapping with uniform prior.

R3.4.3. One recent method (10.1101/2022.01.23.477426) enables genetic variant relevant cell type/state discovery at single-cell resolution by using scATAC-seq. Can the authors apply this approach to their data and compare performance to the mapping methods they have employed?

We thank the reviewer for pointing out this interesting paper. Following the suggestion, we used SCAVENGE to calculate cell enrichment for AF. SCAVENGE computes a trait relevance score (TRS) for each cell for a given trait. In agreement with our results, the high scoring cells are largely cardiomyocytes (Extended Data Figure 8a, Figure 11a below). We also plotted the distribution of TRS scores of different cell types. This clearly shows that the TRS scores for cardiomyocytes are significantly higher than other cell types (Extended Data Figure 8b, Figure 11b below). We have updated the text in the first paragraph of the section in Results, “Open chromatin regions in CMs are enriched with risk variants of heart diseases and inform statistical fine-mapping”. We added:

“We also checked the enrichment of genetic risk of AF at open chromatin regions at individual cells, using the method SCAVENGE (Yu *et al.*, Nat. Biotech., 2020). This analysis confirms that the vast majority of cells enriched with AF risk are CMs (Extended Data Figure 8).”

Figure 11: TRS scores from SCAVENGE are highest in Cardiomyocytes. (a) TRS for each cell plotted on UMAP embedding. (b) Distribution of TRS for each of our cell type clusters.

We’d like to point out that the goal of SCAVENGE is to identify cell types that mediate the genetic effects on a trait. This analysis is similar to our enrichment test using TORUS (Fig. 4a in the manuscript). One advantage of SCAVENGE, compared to ours, is that it does not require explicitly clustering of cells into distinct cell types, and can identify individual cells enriched with trait relevance. This may be beneficial in cases where cells show continuous variations of states (e.g. during differentiation). But in the case of the heart, cells seem to form distinct clusters, so this approach behaves quite similarly to the simpler enrichment analysis we did.

R3.5. Only 6 of 45 genes at $PIP \geq 0.8$ are not the closest genes to the likely causal SNP. Prior studies have shown that nearly half of the target genes are not the closest genes in some cases. Can the authors address these inconsistencies? Ideally, the SNP-target gene pairs should be benchmarked by using Hi-C data or other experimental evidence.

The reviewer made a good point. We believe this is due to the fact that the power of finding high PIP genes is higher in nearest genes than more distal genes. In fact, in calculating gene PIPs, Mapgen gives more weights to variants close to genes than distal variants. For Mapgen to give high PIPs to a distant gene, there must be some additional evidence, such as PC-HiC or ABC, that shows the distal gene is a target of the causal variants. But as we discussed in the paper, often these datasets either do not link variants to any genes; or they link variants to multiple genes, in which case, none of the genes would receive high PIPs. Because of this bias, the genes with high PIPs from Mapgen are more likely to be nearest genes. As such, our results do not contradict the earlier finding that a relatively large fraction of target genes are not nearest ones.

We performed additional benchmark analysis about our discovered genes in this revision. Because our focus is on risk gene discovery, the benchmarking is done at the gene level, instead of the SNP-gene links. We compared Mapgen with other commonly used methods to nominate risk genes. Please see the beginning part of this letter for a description of this benchmark study. Intriguingly, in our evaluation the nearest genes are likely risk genes in about $\frac{1}{2}$ of cases, in agreement with the earlier results cited by the reviewer.

R3.6 The eQTL analyses are interesting and suggest the importance of functional interpretation of variants with single-cell data. However, some statements made appear to be overstated. It is questionable whether bulk-level data is truly underpowered for the discovery of cell-type-specific eQTLs. Prior studies (10.1016/j.ajhg.2018.04.011) have shown that about 12% of 23,000 eQTLs in blood are cell-type-specific. This is also evident in the analysis in Fig. 6b, where CM-related eQTLs showed strong specific activity only in several tissues.

We appreciate the encouraging comment of the reviewer. About the specific point made, we agree with the reviewer that the bulk eQTL data certainly can detect cell-type specific eQTLs. In our results, we found that in heart eQTLs, about 10-15% of them are from cell-type specific OCRs, summing over multiple cell types (Fig. 6d in the submitted manuscript, and Fig. 7d in the revised version). This is roughly in line with the numbers from the paper the reviewer referred to.

When we say the bulk-eQTL studies are under-powered to detect cell-type specific eQTLs, we were not saying that these studies have low power per se, but rather, we were comparing the power of detecting eQTLs acting in a single cell type vs. those acting in multiple cell types. And our results show that the relative power of finding cell-type specific eQTLs is substantially lower. As

a result, among the eQTLs found by these studies, there is an excess of eQTLs shared among multiple cell types, and a relative depletion of eQTLs in single cell types.

This is a subtle point, but we think it has an important implication. This result suggests that bulk eQTLs in general, may miss a large fraction of cell-type specific eQTLs, and that helps explain why the risk variants of complex traits are often not eQTLs. We cited two relevant studies here:

- Chun et al, Limited statistical evidence for shared genetic effects of eQTLs and autoimmune-disease-associated loci in three major immune-cell types, *Nature Genetics*, 2017. This paper showed that in only ~25% loci of autoimmune diseases, the GWAS signals and the eQTLs from immune cells were driven by the same causal variants.
- Yao et al, Quantifying genetic effects on disease mediated by assayed gene expression levels, *Nature Genetics*, 2020. This paper developed a method to quantify how much disease heritability was attributed to eQTLs. Using GTEx eQTL data and a diverse range of traits, it estimated that, averaging across traits, only $11 \pm 2\%$ of heritability was mediated by assayed gene expression levels.

We have revised a large part of this section in response to other reviewers' comments. We rewrote the introduction paragraphs of this section to focus more on the question of why AF variants often do not colocalize with heart eQTLs. Nevertheless, we kept most of the original results in the submitted manuscript.

Minor Comments:

R3.7.1. In Fig. 2a, the data from different donors and from different sources cannot be distinguished in the stacked bar plots. In Line 118 the authors suggest a comparison with another datasets (<https://www.nature.com/articles/s41586-020-2797-4>), but how/where this is shown is not clear.

We omitted these data from the main figure for clarity (we now removed the reference to the different donors/sources from this Figure). However, these info (donor, sampling region) are included in the Extended Data Figure 2.

Previously, we simply used the marker genes for cell types identified in Litvinukova et al. (Litviňuková, M., et al., Cells of the adult human heart, *Nature*, 2020). We now include a direct comparison between our snRNA-seq data and their snRNA-seq data. Using data from Litvinukova et al. as reference we transfer labels to our dataset and observe almost perfect overlap between the cluster assignments. This analysis has also been added as a supplemental figure (Extended Data Figure 3a.). See our response to R3.1 for additional details.

R3.7.2. In Extended Data Fig. 4a,b, the lower boundary is 0.8 and the indicated color is white while there is no white color in the heatmap.

We adjusted the scale so that it does not extend much beyond the spread of the data.

R3.7.3. Lines 143-145. This observation should be expected as the cell-type-specific peaks are used and is unlikely to be due to the fact that myeloid cells are rare.

The reviewer is right that we used cell-type-specific peaks. However, we believe that these observations do indeed suggest that bulk ATAC-seq data is less sensitive to detecting cell-type-specific peaks of constituent cell types if their proportions are small. We rephrased this section (below) and Extended Data Figure 4 provides a qualitative assessment of these conclusions. We also addressed concerns about solely using the proportion of overlaps (see **R3.3**)

“In contrast, the proportions of OCRs from rare cell types (e.g., myeloid) overlapping with bulk DHS was significantly smaller, suggesting that scATAC-seq is more sensitive and detects more regulatory elements in rare cell types compared to bulk DHS (Fig. 3d, top, Extended Data Fig. 5d). This can be seen in several cell-type specific OCRs near some marker genes of rare cell types, which were largely undetected in the pseudo-bulk sample (Extended Data Fig. 4).”

REVIEWERS' COMMENTS

Reviewer #2 (Remarks to the Author):

The revised manuscript from Selewa, et al. nicely addresses my previous major technical concerns. I believe the methodology is novel, sound, and represents an advancement over similar techniques previously developed. I still take some issue, though, with the presentation of some of the results and descriptions of other types of analyses.

Fine-mapping is an analytical technique to provide predictions as to casual variants for complex traits. Each fine mapping technique differs in how they arrive at these predictions, and often some confidence in the prediction based on assumptions in the model and data incorporated can be given. But, I can create a fine-mapping technique that employs me throwing darts at a dartboard. I can use this to make predictions, and I can even assign confidence values. Obviously these will be meaningless, but simply making more high-confidence predictions does not guarantee they are better. That being said, you use other analyses to support your predictions and their increased accuracy over other methods, and I appreciate that. But do not lose sight of the fact that these are predictions, and I'm sure that not every prediction is as highly supported as others.

eQTL analysis (and similarly chromatin interaction analysis) in bulk tissue samples has its limitations. The lack of overlaps between GTEx eQTLs and GWAS loci demonstrate these limitations exist for the sake of identifying causally-associated variants for complex traits. But I would argue it really is not clear why this lack of overlap exists. Being an association model, sample size is important, and so it is likely that adding more samples (potentially a lot more samples) will improve this overlap. eQTLs are also condition specific, and it is possible that performing analysis in healthy tissue from post-mortem donors will miss important disease-associated eQTL. I admit, the mixture of cell types in tissues will also play a role, but it is unclear which of these is having the largest effect, and for a given tissue/disease, each of these may have greater or lesser effects. In this paper, you provide an example of one disease condition and one tissue. In this case, evidence presented here (and elsewhere) suggest that the primary cell type of interest (cardiomyocytes) is one that does not constitute the majority of cells in the associated tissue (heart). Your results, then, can speak to this one case, and it is very plausible that here, the reduced cell fraction is playing a large role in the lack of eQTL identification. But, this is not necessarily generalizable across all tissues, cell types, and traits, nor do you provide evidence that it is. At the end of the day, I would much rather have a strong, tissue-based eQTL that co-localizes with a GWAS locus as evidence of the target gene for that locus than any prediction based on a confluence of more indirect evidence. But I agree, current eQTL analyses are not able to provide this for the majority of GWAS loci. They can for some, though.

I do believe you have presented a method with sufficient evidence that it offers improvements over existing methods and that the predictions are likely strong. But do not oversell what this or any fine mapping method can provide. There is no gold standard set of results to use for evaluation, which means that your methods of evaluation are reasonable, but also means that you have not proven anything about increased accuracy either. The text should reflect this.

Reviewer #4 (Remarks to the Author):

All my concerns have been satisfactorily addressed.

Reviewer 2 provided additional comments on the revised version of our manuscript. The reviewer's comments are copied below in black font with our responses in blue.

The revised manuscript from Selewa, et al. nicely addresses my previous major technical concerns. I believe the methodology is novel, sound, and represents an advancement over similar techniques previously developed. I still take some issue, though, with the presentation of some of the results and descriptions of other types of analyses. Fine-mapping is an analytical technique to provide predictions as to casual variants for complex traits. Each fine mapping technique differs in how they arrive at these predictions, and often some confidence in the prediction based on assumptions in the model and data incorporated can be given. But, I can create a fine-mapping technique that employs me throwing darts at a dartboard. I can use this to make predictions, and I can even assign confidence values. Obviously these will be meaningless, but simply making more high-confidence predictions does not guarantee they are better. That being said, you use other analyses to support your predictions and their increased accuracy over other methods, and I appreciate that. But do not lose sight of the fact that these are predictions, and I'm sure that not every prediction is as highly supported as others. eQTL analysis (and similarly chromatin interaction analysis) in bulk tissue samples has its limitations. The lack of overlaps between GTEx eQTLs and GWAS loci demonstrate these limitations exist for the sake of identifying causally-associated variants for complex traits. But I would argue it really is not clear why this lack of overlap exists. Being an association model, sample size is important, and so it is likely that adding more samples (potentially a lot more samples) will improve this overlap. eQTLs are also condition specific, and it is possible that performing analysis in healthy tissue from post-mortem donors will miss important disease-associated eQTL. I admit, the mixture of cell types in tissues will also play a role, but it is unclear which of these is having the largest effect, and for a given tissue/disease, each of these may have greater or lesser effects. In this paper, you provide an example of one disease condition and one tissue. In this case, evidence presented here (and elsewhere) suggest that the primary cell type of interest (cardiomyocytes) is one that does not constitute the majority of cells in the associated tissue (heart). Your results, then, can speak to this one case, and it is very plausible that here, the reduced cell fraction is playing a large role in the lack of eQTL identification. But, this is not necessarily generalizable across all tissues, cell types, and traits, nor do you provide evidence that it is. At the end of the day, I would much rather have a strong, tissue-based eQTL that co-localizes with a GWAS locus as evidence of the target gene for that locus than any prediction based on a confluence of more indirect evidence. But I agree, current eQTL analyses are not able to provide this for the majority of GWAS loci. They can for some, though. I do believe you have presented a method with sufficient evidence that it offers improvements over existing methods and that the predictions are likely strong. But do not oversell what this or any fine mapping method can provide. There is no gold standard set of results to use for evaluation, which means that your methods of evaluation are reasonable, but also means that you have not proven anything about increased accuracy either. The text should reflect this.

We thank the reviewer for the positive evaluation of our revised manuscript and appreciate the comments regarding the limitations of fine-mapping and the overlap between eQTLs and GWAS SNPs.

We agree that these are important points and we included them in our discussion of the limitations of our approach. The re-written section is provided below:

Despite the advances described above, our study has a few limitations. Our experimental data were limited to four anatomical locations of the ventricles, while some AF risk variants might act through atrial-specific CMs. However, it is worth noting that a recent study, using scRNA-seq based cellular atlas of the heart including all anatomic locations, found that AF candidate genes were strongly enriched in ventricular CMs (pmid: 32971526). Additionally, our data were from adult hearts, and thus may miss regulatory elements acting transiently during development. Our computational procedure relied on statistical fine-mapping, which may provide mis-calibrated results in practice (pmid: 36643910). To prioritize genes, we used a set of heuristic rules to link variants to genes. This worked reasonably well in our data, but without comprehensive evaluation it is difficult to know how well it will perform in other settings. This challenge is exacerbated by the lack of a gold standard dataset to evaluate risk genes from GWAS. Lastly, we showed that bulk eQTLs largely missed the effects of our fine-mapped variants and our analysis suggested that one factor may be the reduced power of detecting cell type specific regulatory effects. We cannot, however, exclude other explanations. For example, many variants may act during cardiomyocyte differentiation and therefore would not be detected as eQTLs in adult samples. We believe future eQTL studies across multiple cell types and different developmental stages would help bridge the gap of our understanding.